# Does "Do Differentiable Simulators Give Better Policy Gradients?" Give Better Policy Gradients?

**Ku Onoda    Paavo Parmas    Manato Yaguchi    Yutaka Matsuo**
The University of Tokyo, Tokyo, Japan
{ku.onoda, paavo.parmas, manato.yaguchi, matsuo}@weblab.t.u-tokyo.ac.jp

## Abstract

In policy gradient reinforcement learning, access to a differentiable model enables 1st-order gradient estimation that accelerates learning compared to relying solely on derivative-free 0th-order estimators. However, discontinuous dynamics cause bias and undermine the effectiveness of 1st-order estimators. Prior work addressed this bias by constructing a confidence interval around the REINFORCE 0th-order gradient estimator and using these bounds to detect discontinuities. However, the REINFORCE estimator is notoriously noisy, and we find that this method requires task-specific hyperparameter tuning and has low sample efficiency. This paper asks whether such bias is the primary obstacle and what minimal fixes suffice. First, we re-examine standard discontinuous settings from prior work and introduce DDCG, a lightweight test that switches estimators in nonsmooth regions; with a single hyperparameter, DDCG achieves robust performance and remains reliable with small samples. Second, on differentiable robotics control tasks, we present IVW-H, a per-step inverse-variance implementation that stabilizes variance without explicit discontinuity detection and yields strong results. Together, these findings indicate that while estimator switching improves robustness in controlled studies, careful variance control often dominates in practical deployments.

## 1 Introduction

Policy gradient methods seek to optimize a parameterized policy $\boldsymbol{\theta}$ by estimating the gradient of the expected return, $\hat{\boldsymbol{g}} \approx \frac{\mathrm{d}}{\mathrm{d}\boldsymbol{\theta}} \mathbb{E}_{p(\boldsymbol{\tau})}\left[R(\boldsymbol{\tau})\right]$. In the most general setting—where the environment is treated as a black box—0th-order estimators such as REINFORCE (Williams, 1992) are often used. While broadly applicable, these estimators suffer from high variance and poor sample efficiency. When a differentiable simulator is available, 1st-order gradient estimators (e.g., via the reparameterization trick (Kingma et al., 2015)) can substantially reduce variance and accelerate convergence. However, real-world systems often involve contacts, friction, or other nonsmooth effects, producing discontinuities that bias 1st-order estimates (Lee et al., 2018; Parmas and Sugiyama, 2021).

Each approach has its own advantages and disadvantages, and one way to leverage their strengths is by mixing the estimators (Parmas et al., 2018; 2023). Parmas et al. motivate this composite view by showing that chaotic dynamics can cause gradient variance to explode with horizon, and advocate inverse variance weighting (IVW) as a principled way to set the mixture weights. Specifically, these methods compute

$$\hat{\boldsymbol{g}} = \alpha\hat{\boldsymbol{g}}_1 + (1 - \alpha)\hat{\boldsymbol{g}}_0,$$

where $\hat{\boldsymbol{g}}_0$ and $\hat{\boldsymbol{g}}_1$ are 0th- and 1st-order estimates respectively, and $\alpha \in [0, 1]$ is a weighting parameter. Under IVW, $\alpha = \frac{\mathbb{V}[\hat{\boldsymbol{g}}_0]}{\mathbb{V}[\hat{\boldsymbol{g}}_0] + \mathbb{V}[\hat{\boldsymbol{g}}_1]}$. When the variances are estimated accurately, IVW can improve performance by reducing *variance*. Despite its appeal, IVW may fail in domains with discontinuities or contact dynamics. As the work of Suh et al. (2022) shows, sharp changes in the reward landscape create situations where the 1st-order gradient exhibits large errors but spuriously shows low empirical variance in finite samples. This phenomenon, called "*empirical bias*," leads IVW to overweight corrupt 1st-order estimates, harming performance. Building on the same $\alpha$-mixing scheme above, Suh

et al. (2022) propose alpha-order batched gradient (AoBG), which augments IVW with a discontinuity-detection rule based on confidence intervals around the REINFORCE estimator. However, since REINFORCE can be extremely noisy, these intervals are broad, reducing sample efficiency and necessitating extensive task-specific parameter tuning. Moreover, while prior reports discuss AoBG behavior, they do not establish robust success on standard robotic control benchmarks (Gao et al., 2024).

In this paper, we pursue two concrete goals focused on reassessing composite gradient methods and examining whether the *variance* or *empirical bias* is the main obstacle to practical performance.

First, we *reassess existing work and expose fundamental shortcomings*. Specifically, we re-establish the existence of the finite-sample bias phenomenon—where 1st-order gradients can appear low-variance yet be inaccurate—and introduce *Discontinuity Detection Composite Gradient (DDCG)*, which uses a lightweight statistical test to decide when to trust 1st-order information. We reproduce and re-evaluate all experiments from the AoBG paper under the same settings (Suh et al., 2022) and show that DDCG achieves results comparable to or better than AoBG, with substantially improved robustness to hyperparameters and reliable behavior in small-sample regimes.

Second, we ask whether this bias is actually the primary obstacle in practical robotics control. Prior studies (Son et al., 2023; Gao et al., 2024) reported limited performance or incomplete realizations of inverse-variance mixing; we therefore provide a clear, per–time-step implementation, *IVW-H*, to isolate the role of variance control in practice. On standard robotics tasks, IVW-H attains strong performance without explicit discontinuity detection, suggesting that stabilizing variance at the step level can be sufficient, while the role of *empirical bias* appears minimal in these settings.

## 2 RELATED WORK

**Differentiable Simulators.** Recent advances in differentiable simulators enable gradient-based policy optimization with either automatic differentiation (Griewank and Walther, 2003; Heiden et al., 2021; Freeman et al., 2021) or analytic derivatives (Carpentier and Mansard, 2018; Geilinger et al., 2020; Werling et al., 2021). These methods reduce variance in gradient estimates and often accelerate learning. However, contact-rich or discontinuous dynamics remain challenging because the inherent nonsmoothness introduces bias or instability in 1st-order gradient estimates, undermining their reliability for optimization tasks.

**Composite Gradient Estimators.** Combining 0th-order and 1st-order gradients can balance robustness and efficiency. Parmas et al. (2018) propose Total Propagation (TP), which uses inverse variance weighting (IVW) to mix gradients. However, discontinuities can introduce biased 1st-order gradients (Lee et al., 2018; Parmas and Sugiyama, 2021), and IVW can fail when these biases are underestimated. Suh et al. (2022) address this "empirical bias" phenomenon by a scheme that constructs confidence intervals around 0th-order gradient estimates to detect bias.

**Policy Optimization with Differentiable Simulation.** Analytic Policy Gradient (APG) (Freeman et al., 2021) computes policy gradients directly from simulator-provided derivatives, accelerating learning but not explicitly addressing discontinuities. Short-Horizon Actor-Critic (SHAC) (Xu et al., 2022) reduces variance by truncating rollouts and using a terminal value to smooth the objective, enabling effective use of analytic gradients. Adaptive-Gradient Policy Optimization (AGPO) (Gao et al., 2024) mitigates nonsmoothness by adapting weights based on batch-gradient variance, while Gradient-Informed PPO (GIPPO) (Son et al., 2023) introduces an $\alpha$-policy that downweights unreliable analytic gradients within a PPO framework.

## 3 BACKGROUND

**Notation.** Throughout this paper, we use bold font (e.g., $\boldsymbol{x}$) to represent tensors unless otherwise stated. Here, $\hat{\mathbb{E}}$ denotes the sample mean of the corresponding quantity. We define the empirical variance of a set of $N$ samples as

$$\hat{\mathbb{V}}[\cdot] = \frac{1}{N-1} \sum_{i=1}^{N} \left( (\cdot)_i - \hat{\mathbb{E}}[\cdot] \right)^2.$$

**Task setting.** We consider finite horizon control tasks with state variables $s$, and actions $a$ that are computed from a policy $\pi_\zeta$. States transition according to the dynamics $p(s'|s, a)$; following actions according to the policy $\pi_\zeta$ leads to trajectories $\tau_\zeta = (s_0, a_0, s_1, \ldots, s_H)$. We consider the objective $\mathbb{E}[R(\tau_\zeta)]$, where $R(\tau_\zeta)$ is a cumulative sum of scalar rewards computed by the reward function $r(s, a)$. We aim to maximize this objective using gradient ascent.

**Bias-Variance Error Decomposition.** A central theme in estimating gradients or any statistical inference is the interplay between bias and variance. For an estimator $\hat{Z}$ of $Z$, the mean squared error (MSE) can be expressed as

$$\underbrace{\mathbb{E}\left[(\hat{Z} - Z)^2\right]}_{\text{Error}} = \underbrace{\left(\mathbb{E}[\hat{Z}] - Z\right)^2}_{\text{Bias}} + \underbrace{\mathbb{E}\left[(\hat{Z} - \mathbb{E}[\hat{Z}])^2\right]}_{\text{Variance}}. \tag{1}$$

An estimator is unbiased if $\mathbb{E}[\hat{Z}] = Z$. In gradient-based methods, a low-bias estimator may still exhibit high variance, hindering learning efficiency. Conversely, reducing variance may introduce systematic bias. Balancing bias and variance is therefore a key challenge in designing gradient estimators, motivating strategies to control variance without incurring significant bias.

**Elementary Gradient Estimators.** We perform randomized smoothing and sample policy parameters $\zeta \sim p(\zeta; \theta)$. Let $\theta$ denote the parameters to be optimized, and let $\tau_\zeta$ represent a random trajectory or episode whose distribution depends on $\theta$. In particular, in the current work $p(\zeta; \theta)$ will always be Gaussian, with $\theta$ as the mean of this Gaussian. That is, we can write

$$\zeta = \theta + \sigma\,\epsilon, \quad \epsilon \sim \mathcal{N}(0, I). \tag{2}$$

A gradient estimator $\hat{g}$ is unbiased if

$$\mathbb{E}[\hat{g}] = \frac{\mathrm{d}}{\mathrm{d}\theta}\mathbb{E}_{p(\tau_\zeta; \theta)}\left[R(\tau_\zeta)\right]. \tag{3}$$

Here, "unbiased" refers to the estimator being unbiased for the gradient of the objective inside the simulation model (Eq. 3). Transfer errors from simulation to the real world are a separate concern addressed by sim-to-real techniques.

**0th-order estimator.** A widely used unbiased method is the score function or likelihood ratio approach (Glynn, 1990), often referred to as REINFORCE (Williams, 1992). It can be written as

$$\hat{g}_0(\theta) = \frac{\mathrm{d}\log p(\tau; \theta)}{\mathrm{d}\theta}\left(R(\tau) - b\right), \tag{4}$$

where $\tau$ represents a sample from $p(\tau; \theta)$, and $b$ is a baseline that can reduce variance (Berahas et al., 2022). In our experiments, we follow Suh et al. (2022) and use a deterministic baseline given by the objective evaluated at the mean parameter, $b = f(\theta) = \mathbb{E}_{p(\tau_\zeta; \theta)}[R(\tau_\zeta)]$, which does not depend on the Monte Carlo samples used in the gradient estimate and thus keeps the estimator unbiased. Despite being unbiased, this estimator often suffers from high variance, which can significantly increase the number of samples required for effective learning.

**1st-order estimator.** An alternative approach, known as the reparameterization trick (Kingma et al., 2015) or pathwise derivative (Schulman et al., 2015), avoids directly differentiating through a probability distribution by defining a deterministic transformation

$$\tau = \mathcal{T}_\theta(\epsilon), \quad \epsilon \sim p(\epsilon). \tag{5}$$

Because $\tau$ still has distribution $p(\tau; \theta)$ by construction, one obtains the 1st-order estimator:

$$\hat{g}_1(\theta) = \frac{\mathrm{d}R}{\mathrm{d}\tau}\frac{\mathrm{d}\mathcal{T}_\theta(\epsilon)}{\mathrm{d}\theta}. \tag{6}$$

This estimator remains unbiased if $R$ is continuous, and in practice, it often exhibits lower variance than $\hat{g}_0$. Consequently, it tends to be more sample-efficient for continuous parameter and action spaces. For instance, if we reparameterize $\zeta$ as in Eq. (2), then $\frac{\partial\zeta}{\partial\theta} = I$ and $\frac{\partial\zeta}{\partial\sigma} = \epsilon$, which simplifies $\hat{g}_1$ to $\frac{\mathrm{d}R}{\mathrm{d}\zeta}$ with respect to $\theta$. However, when $R$ is discontinuous, the 1st-order estimator can be biased.

**Batch Estimation.** While Eq. (4) and Eq. (6) are presented as single-sample estimators for clarity, in practice (and in our experiments), these are estimated using a batch of $N$ samples to increase accuracy, and compute empirical means and variances.

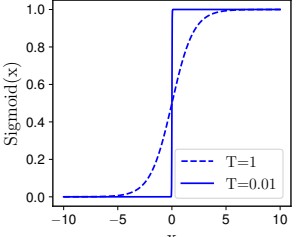 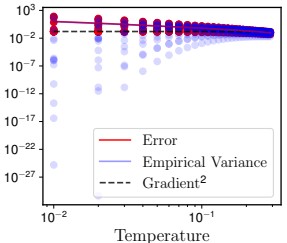

(a) Discontinuous-like behavior    (b) Error, Empirical Variance

Figure 1: Sigmoid Function

**Composite Gradient Estimators.** Although the 1st-order estimator $\hat{g}_1$ typically has lower variance than the 0th-order $\hat{g}_0$, it may be biased in the presence of discontinuities. A practical approach by Parmas et al. (2018) mixes these estimators via a linear combination:

$$\hat{g}_\alpha = \alpha\hat{g}_1 + (1-\alpha)\hat{g}_0, \quad \alpha \in [0,1], \tag{7}$$

where $\alpha$ close to 1 emphasizes the 1st-order estimator while $\alpha$ near 0 relies more on the 0th-order method. Additionally, they propose leveraging Inverse Variance Weighting (IVW) to optimally select $\alpha$ in their Total Propagation (TP) framework. Under the simplifying assumption that $\hat{g}_0$ and $\hat{g}_1$ are uncorrelated, the theoretically optimal weight $\alpha_{\mathrm{opt}}$ that minimizes the variance of $\hat{g}_\alpha$ is

$$\alpha_{\mathrm{opt}} = \frac{\mathbb{V}[\hat{g}_0]}{\mathbb{V}[\hat{g}_0] + \mathbb{V}[\hat{g}_1]}. \tag{8}$$

If the covariance between $\hat{g}_0$ and $\hat{g}_1$ is non-negligible, the above expression must be adjusted accordingly, as discussed in (Parmas et al., 2023). Nevertheless, in the uncorrelated case,

$$\frac{1}{\mathbb{V}[\hat{g}_\alpha]} = \frac{1}{\mathbb{V}[\hat{g}_0]} + \frac{1}{\mathbb{V}[\hat{g}_1]}, \tag{9}$$

indicating that the combined estimator can achieve a strictly lower variance than either $\hat{g}_0$ or $\hat{g}_1$ alone.

In practice, the true variances $\mathbb{V}[\hat{g}_0]$ and $\mathbb{V}[\hat{g}_1]$ are generally unknown and must be approximated from sample data. Explicitly, one computes $\hat{\mathbb{V}}[\hat{g}_0]$ and $\hat{\mathbb{V}}[\hat{g}_1]$ to obtain $\hat{\alpha}_{\mathrm{opt}}$. This creates difficulties whenever the empirical variance estimates are poor, notably in discontinuous environments.

**Limitations of Empirical Variance Estimation** While IVW often performs well, Suh et al. (2022) points out that certain practical factors—such as contact, friction, or discontinuities in physics simulations—can cause an "empirical bias" phenomenon, resulting in gradients that exhibit low empirical variance yet remain highly inaccurate. An illustrative example involves the Sigmoid function, $\mathrm{Sigmoid}(x) = \frac{1}{1+\exp\left(-\frac{x}{T}\right)}$. As shown in Figure 1a, when the temperature $T$ is large, the function is fairly smooth. However, at very small $T$, it transitions sharply and resembles a discontinuity. Although $\mathrm{Sigmoid}(x)$ is mathematically continuous for any finite $T$, its narrow transition region makes finite-sample gradient estimates prone to large, sporadic errors.

From the perspective of the bias-variance decomposition Eq. (1), an unbiased estimator's error coincides exactly with its variance (since Bias $= 0$). In principle, this means that the true variance of the Sigmoid gradient should match the observed error. However, as Figure 1b shows, the empirical variance computed from a small batch often fails to reflect the true error. The reason is that very large gradients occur with small probability, causing the true variance to be very large (sometimes viewed as "infinite variance" in the limit of vanishing probability). A mathematical example illustrating how this "infinite variance" phenomenon arises is given in Appendix B. In practice, a finite sample may overlook those rare but significant gradients, leading to a systematic underestimation of the variance. This phenomenon underscores a fundamental challenge: when an unbiased gradient estimator has heavy-tailed or rare large-magnitude events, the empirical variance can severely underestimate the true variance.

**Interpolation Protocol (AoBG)** The AoBG method proposed by Suh et al. (2022) builds upon the IVW framework by introducing additional safeguards against discontinuities. AoBG starts with $\alpha_{\mathrm{opt}}$ but modifies it based on a measure of potential bias $B = \|\hat{g}_1 - \hat{g}_0\|_2$:

$$\alpha_\gamma := \begin{cases} \alpha_{\mathrm{opt}} & \text{if } \alpha_{\mathrm{opt}}B \leq \gamma - \varepsilon, \\ \frac{\gamma-\varepsilon}{B} & \text{otherwise.} \end{cases} \tag{10}$$

This formulation introduces a precision threshold $\gamma$ to control acceptable bias and a confidence term $\varepsilon$ to account for uncertainty in the 0th-order estimator. When potential bias is too large, the method reduces $\alpha$ to maintain precision, effectively reverting to the reliable 0th-order estimator in challenging areas. For small sample sizes, a conservative approach uses only the 0th-order gradient ($\alpha = 0$), though this raises several concerns.

First, with small sample sizes, the 1st-order estimator is typically more effective due to its lower variance. Thus, relying on the 0th-order gradient seems counterintuitive, potentially leading to suboptimal outcomes. Second, selecting the parameter $\gamma$ for each task requires task-specific tuning, limiting the method's generalizability and usability. Eliminating the need for such parameter adjustments would make the method more robust and practical across diverse scenarios.

## 4    Proposal

### 4.1    Discontinuity Detection Composite Gradient (DDCG)

We propose *Discontinuity Detection Composite Gradient (DDCG)*, which keeps the usual inverse-variance mix of 0th- and 1st-order estimators but *gates* the use of the 1st-order term by a simple statistical test. The gate is derived from two standard conditions under which IVW is trustworthy:

- **(A1) Reliable variance:** the empirical variance of the 1st-order gradient is close to its true variance (so IVW weights are meaningful).
- **(A2) Local smoothness:** $f$ is locally well-behaved (e.g., near-quadratic), making the 1st-order gradient accurate and low-variance (Xu et al., 2019; Domke, 2019).

If (A1) holds, IVW already downweights noisy 1st-order terms; but (A2) is also needed to avoid using biased 1st-order estimates near discontinuities. We therefore run a statistical test that passes with probability at least $1 - \delta$ when (A1) and (A2) hold; if it passes we apply IVW, otherwise we fall back to the 0th-order estimator. Importantly, these assumptions are *not* required for the algorithm to run: they are only checked to decide whether to trust IVW.

**Step 1: Variance Reliability**   The first (A1) concerns the accuracy of the empirical variance estimate of 1st-order gradients. If this assumption holds, we can rely on the sample-based variance used by IVW to be close to the true variance.

Formally, suppose we have $N$ samples $\{\boldsymbol{x}_i\}_{i=1}^N$ from a function $f$, along with their function values $\{f(\boldsymbol{x}_i)\}_{i=1}^N$ and gradients $\{\nabla f(\boldsymbol{x}_i)\}_{i=1}^N$. Denote:

$$\hat{\boldsymbol{v}} \;=\; \frac{1}{N-1} \sum_{i=1}^N \big\| \nabla f(\boldsymbol{x}_i) \;-\; \overline{\nabla f} \big\|_2^2, \tag{11}$$

where $\overline{\nabla f} = \frac{1}{N} \sum_{i=1}^N \nabla f(\boldsymbol{x}_i)$ is the empirical mean of the gradients. We assume that $\hat{\boldsymbol{v}}$ differs from the true variance of $\nabla f(\boldsymbol{x})$ by at most $\varepsilon_v$:

$$\Big| \, \hat{\boldsymbol{v}} \;-\; \mathbb{E}_{\boldsymbol{x}}\big[\|\nabla f(\boldsymbol{x}) - \mathbb{E}_{\boldsymbol{y}}[\nabla f(\boldsymbol{y})]\|_2^2\big] \Big| \;\leq\; \varepsilon_v. \tag{12}$$

Such a bound can be derived via standard statistical results (e.g., chi-squared-based confidence intervals). By enforcing a maximal floor on $\hat{\boldsymbol{v}}$, we reduce the risk of underestimating gradient variance, and thus overweighting a potentially high-variance 1st-order estimator.

**Step 2: Discontinuity Detection**   To derive the statistical test, we assume that $f$ is sufficiently smooth so that 1st-order gradients remain accurate. In practice, smoothness ensures that the variance of 1st-order estimates does not explode.

To merge (A1) and (A2) into a single test, we assume a Lipschitz-like condition on gradient changes:

$$\|\nabla f(\boldsymbol{x}) - \nabla f(\boldsymbol{y})\| \approx L\|\boldsymbol{x} - \boldsymbol{y}\|, \tag{13}$$

where $L$ is a local curvature constant. We then compare the variance of a quadratic approximation of $f$ with the empirical gradient variance. Under smoothness and bounded residuals, a condition emerges (detailed in Appendix C):

$$\hat{\boldsymbol{v}} + \varepsilon_v \;\overset{?}{\geq}\; 2(1-c)\frac{\mathbb{V}\left[f(\boldsymbol{x})\right]}{\sigma^2} - 2\|\overline{\nabla f}\|^2, \tag{14}$$

Here the right-hand side corresponds to the gradient variance that a locally quadratic model of $f$ would induce under randomized smoothing, while the term involving $\|\overline{\nabla} f\|^2$ subtracts the contribution of the mean gradient. Intuitively, if $f$ is smooth and our variance estimates are reliable, the empirical gradient variance cannot be much *smaller* than this quadratic proxy: large fluctuations in function values necessarily imply non-trivial fluctuations in the gradient. When the left-hand side falls below the right-hand side, we interpret this as evidence of heavy-tailed or discontinuous behavior that makes the IVW weights unreliable, and we fall back to the 0th-order estimator.

In implementation, all quantities in Eq. (14) are computed from the same finite batch of samples: we replace $\mathbb{V}[f(\boldsymbol{x})]$ by its empirical counterpart $\hat{\mathbb{V}}[f(\boldsymbol{x})]$. Analogously to Eq. (12), one could also attach an explicit confidence interval to this scalar variance estimate; we found that doing so did not materially change the decisions of the test, so for simplicity we omit this extra term in the main algorithm.

**Interpretation of c.** In Eq. (14), the parameter $c$ relaxes the requirement that $f$ be perfectly quadratic. If $f$ were exactly quadratic, then taking $c = 0$ would make the inequality tight in that ideal case. As $c$ increases above 0, we allow more deviation of $f$ from perfect quadratic behavior, permitting greater nonlinearity or mild discontinuities. Thus, a smaller $c$ imposes stricter smoothness requirements on $f$, while a larger $c$ offers more flexibility for $f$ to deviate from a purely quadratic shape. In our experiments we simply fix $c = 0.3$ across all AoBG benchmarks, and Appendix H shows that DDCG is robust for any $c \in [0.1, 0.9]$ on all considered tasks.

**Step 3: Adaptive Weighting** Given the test in Eq. (14), we define the composite gradient estimator by adaptively selecting weight $\alpha$ between 0th- and 1st-order estimators:

$$\hat{\alpha} := \begin{cases} \hat{\alpha}_{\text{opt}} & \text{if Eq. (14) holds,} \\ 0 & \text{otherwise.} \end{cases} \tag{15}$$

Here, $\hat{\alpha}_{\text{opt}}$ is the inverse-variance-optimal weight computed from the empirical variances of the 0th-order and 1st-order gradients. In other words: If the test passes, we assume that (A1) and (A2) both hold and can therefore exploit the lower variance of the 1st-order estimator through IVW. If the test fails, we set $\alpha = 0$, reverting to a purely 0th-order estimator to avoid biased 1st-order gradients.

**Summary.** DDCG utilizes the 1st-order gradient's lower variance wherever it is safe to do so. Our two assumptions—(A1) accurate empirical variance estimation and (A2) local smoothness—ensure that IVW is likely reliable. By checking Eq. (14), we detect plausible violations of either assumption. Failing this test triggers a fallback to safe 0th-order methods. In practice, this mechanism obviates the need for extensive hyperparameter tuning; aside from $\delta$ (which controls confidence) and $c$ (which bounds how non-quadratic the function may be), the method remains largely automatic.

**Comparison with AoBG.** Our DDCG method and AoBG share the idea of constructing a statistical estimator for bias; however, a crucial difference is that AoBG uses the $\frac{\mathrm{d}\log p(\tau;\theta)}{\mathrm{d}\theta}$ terms in the notoriously noisy REINFORCE estimator to construct a confidence interval. In contrast, our estimator in Eq. (14) uses only the function value and gradient variances. Consequently, in motivational toy tasks, the estimation of our bounds is $d$ times more efficient than that of AoBG (Appendix D), where $d$ denotes the number of dimensions.

## 4.2 Stepwise Inverse Variance Weighting (IVW-H)

We adopt a *stepwise* (per–step, per–action) inverse-variance weighting scheme. Let $t \in \{0, \ldots, H - 1\}$ index time steps, $n$ index actors (parallel rollouts), and let bold symbols denote action-dimensional vectors in $\mathbb{R}^A$. For each $(t, n)$, let $\hat{\boldsymbol{g}}_{0,t,n}$ and $\hat{\boldsymbol{g}}_{1,t,n}$ be the 0th- and 1st-order gradient vectors. We estimate empirical variances across actors at fixed $t$ elementwise,

$$\hat{\boldsymbol{v}}_{0,t,a} = \hat{\mathbb{V}}_n[\hat{\boldsymbol{g}}_{0,t,n,a}], \qquad \hat{\boldsymbol{v}}_{1,t,a} = \hat{\mathbb{V}}_n[\hat{\boldsymbol{g}}_{1,t,n,a}]. \tag{16}$$

IVW-H assigns a per-step, per-dimension IVW weight

$$\hat{\alpha}_{t,a} = \frac{\hat{\boldsymbol{v}}_{0,t,a}}{\hat{\boldsymbol{v}}_{0,t,a} + \hat{\boldsymbol{v}}_{1,t,a}} \in [0, 1], \tag{17}$$

and forms the combined gradient elementwise as

$$\hat{\boldsymbol{g}}_{\alpha,t,n,a} = \hat{\alpha}_{t,a} \hat{\boldsymbol{g}}_{1,t,n,a} + (1 - \hat{\alpha}_{t,a}) \hat{\boldsymbol{g}}_{0,t,n,a}. \tag{18}$$

The combination is applied elementwise over $(t, n, a)$ and then backpropagated through the policy network parameters. Following prior practice in total propagation-style estimators (Parmas, 2020), variance across actors at fixed $(t, a)$ yields an efficient and stable estimate that aligns with per-step aggregation in trajectory optimization. This action-space formulation mirrors the Total Propagation X algorithm (Parmas et al., 2023): we first form a composite gradient with respect to actions and then backpropagate it through the policy network. In principle, a full TPX implementation should be the stronger estimator, and we expect it to outperform IVW-H when it can be implemented efficiently; however, TPX can be cumbersome to realize in practice due to simulator-specific implementation details. For reference, TPX is implemented in the Proppo framework (Parmas and Seno, 2022). Crucially, the batched actors at each time step provide the sample dimension needed for the empirical variances without introducing extra simulator calls, so the wall-clock cost of IVW-H is comparable to that of a pure 1st-order baseline. This stands in contrast to parameter-space IVW implementations such as the one reported in GIPPO (Son et al., 2023), which require additional simulator evaluations and were observed to be much slower and less effective. The pseudocode of the algorithm is provided in Appendix E.

## 5 EXPERIMENTS

### 5.1 OVERVIEW

We pursue two goals: (i) re-examine AoBG in explicit empirical-bias settings and evaluate DDCG in the same regimes; (ii) test whether variance—not bias—is the practical bottleneck on standard continuous-control benchmarks via IVW-H. Unless otherwise noted, all curves are averaged over multiple random seeds; the number of "trials" reported in Appendix K coincides with the number of seeds for each setting.

**Part I: Empirical-bias regimes (re-evaluating AoBG and validating DDCG).** We revisit settings where empirical bias is known to arise and analyze AoBG's behavior (including the trajectory of the weighting parameter $\alpha$) alongside IVW and baselines. We then evaluate whether DDCG improves outcomes under the same conditions. Following the original setup, we compare five approaches: 0th-order grad, 1st-order grad, AoBG (parameter $\gamma$), IVW, and DDCG (parameter $c$ with statistical test confidence $\delta = 0.05$). Unless otherwise noted, DDCG uses a unified hyperparameter $c = 0.3$; sensitivity is reported in Appendix H. The function-optimization (toy) experiments supporting the landscape analysis and $\alpha$-selection diagnostics are in Appendix F.

**Part II: Practical continuous control (IVW-H).** To probe whether empirical bias is the primary issue in practical settings, we conduct experiments in differentiable physics with MuJoCo-style tasks (*CartPole*, *Hopper*, *Ant*), following prior usage in GIPPO and SHAC. In these domains, we hypothesize that explicit discontinuity detection is largely unnecessary because standard variance weighting (IVW) is already sufficient. Indeed, our sensitivity analysis (see Figure 16 in Appendix I) reveals that AoBG does not outperform simple IVW, implying that empirical bias is not the dominant bottleneck. Therefore, rather than applying DDCG, we focus on demonstrating that IVW-H—a simple, computationally efficient stepwise update—is sufficient to outperform complex state-of-the-art baselines like GIPPO. We compare 1st-order grad, 0th-order grad, AoBG, IVW, IVW-H (our per-step, per-action IVW), and GIPPO.

### 5.2 DIFFERENTIABLE SIMULATION TASKS

First, we examine tasks that model physical systems with contact and friction. The setup was replicated using Suh (2021)'s code, implemented in Suh et al. (2022), enabling direct comparison. Tasks fall into three categories: Landscape Analysis, Trajectory Optimization, and Policy Optimization. AoBG relies on tuned parameters; DDCG fixes $c = 0.3$. Since their paper lacks specifics, we set AoBG to the default parameters in the code. Refer to Appendix K for detailed parameter settings.

#### 5.2.1 LANDSCAPE ANALYSIS

**Experimental Setup.** We study discontinuous landscapes to quantify estimation error and $\alpha$ selection, and we perform landscape optimization while visualizing cost convergence and $\alpha$ for AoBG, IVW, and DDCG (the $\alpha$ visualization and IVW comparison were not included in Suh et al. (2022)). We use

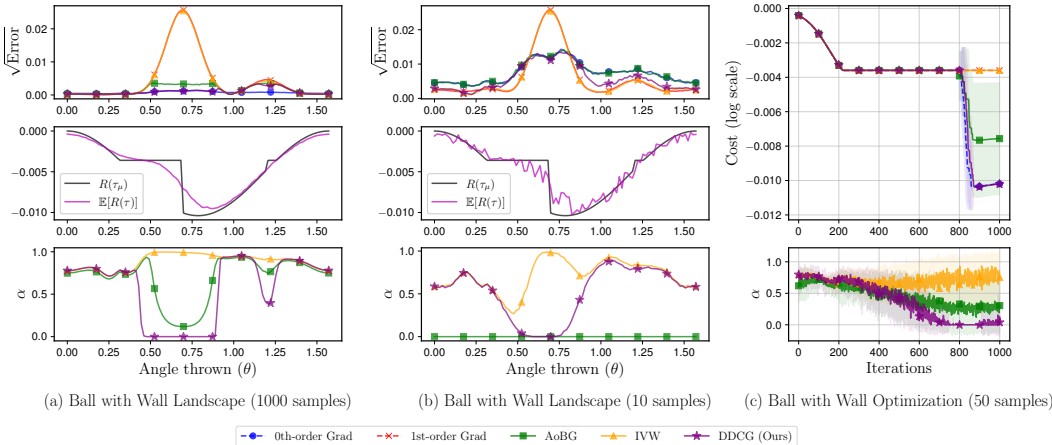

(a) Ball with Wall Landscape (1000 samples)    (b) Ball with Wall Landscape (10 samples)    (c) Ball with Wall Optimization (50 samples)

Figure 2: Ball with Wall. Columns 1, 2: top row shows the square root of estimation errors (scaled to match the previous study), middle row shows the cost function, and bottom row shows $\alpha$ selection. Column 3: optimization cost and $\alpha$ selection.

two tasks that exhibit collision-induced discontinuities: *Ball with Wall* (maximize travel distance with impacts) and *Momentum Transfer* (maximize angular momentum transfer). For brevity we report *Ball with Wall* in the main text and defer *Momentum Transfer* to Appendix G. Both tasks follow the setup of Suh et al. (2022) for fair comparison with AoBG.

**Findings.** For larger sample sizes ($N = 1000$) in Figure 2(a), IVW remains biased near collisions due to an overconfident 1st-order component. Both AoBG ($\gamma = 0.005$) and DDCG detect these discontinuities and reduce the weighting parameter $\alpha$. For smaller sample sizes ($N = 10$), Figure 2(b) shows that AoBG's fixed $\gamma$ becomes overly conservative, with $\alpha$ dropping to zero, underutilizing available gradient information. In contrast, DDCG continues to detect discontinuities robustly using the same parameters. The cost convergence in Figure 2(c) confirms that both AoBG and DDCG avoid collisions by shifting toward the 0th-order estimator. Similar trends are observed in the Momentum Transfer task. A detailed analysis—including the variance and bias of the estimators, as well as complete results for Momentum Transfer—is provided in Appendix G.

### 5.2.2 TRAJECTORY OPTIMIZATION

In trajectory optimization, a sequence of control inputs is optimized for a known environment and initial conditions. We evaluated two tasks, Pushing and Friction, where contact and friction can make 1st-order gradients inaccurate.

**Pushing.** Two rigid bodies collide with varying spring constants $k$: a smaller $k$ results in "soft" collisions, while a larger $k$ leads to "stiff" ones. We apply force to the first body to minimize the second body's distance to the destination. AoBG was tuned per stiffness. For soft collisions, $\gamma = 1000$ (the original parameter was extremely large and effectively loosened the constraint so that AoBG always used IVW, so we used a smaller value). For stiff collisions, we set $\gamma = 10^8$. For DDCG, $c = 0.3$. Figure 3(a) and (b) show that under soft collisions, both AoBG and DDCG favor 1st-order gradients. In the low-sample setting (Figure 3(b)), AoBG conservatively relies on the 0th-order component, failing to leverage faster 1st-order convergence, while DDCG continues using 1st-order gradients. Under stiff collisions, shown in Figure 3(c), we initially expected first-order gradients to be biased. However, we see that both AoBG and DDCG optimized by selecting $\alpha$ values near IVW indicating that the stiffness caused variance instead.

**Friction.** Two overlapping objects interact under Coulomb friction, where static and dynamic friction cause abrupt transitions at near-zero relative velocity. A force is applied to object 1 to move object 2 toward the goal. In the original code, AoBG was not properly tuned, preventing effective use of 1st-order estimator; consequently, we re-tuned AoBG ($\gamma = 30000$). Furthermore, to enable clearer sample comparisons, we assumed a larger sample size than in the code ($N = 100$). For DDCG, we set $c = 0.3$. Figure 4(a) and (b) show that the 1st-order estimator and IVW stall once friction thresholds are crossed. AoBG and DDCG detect and mitigate these discontinuities by shifting more weight

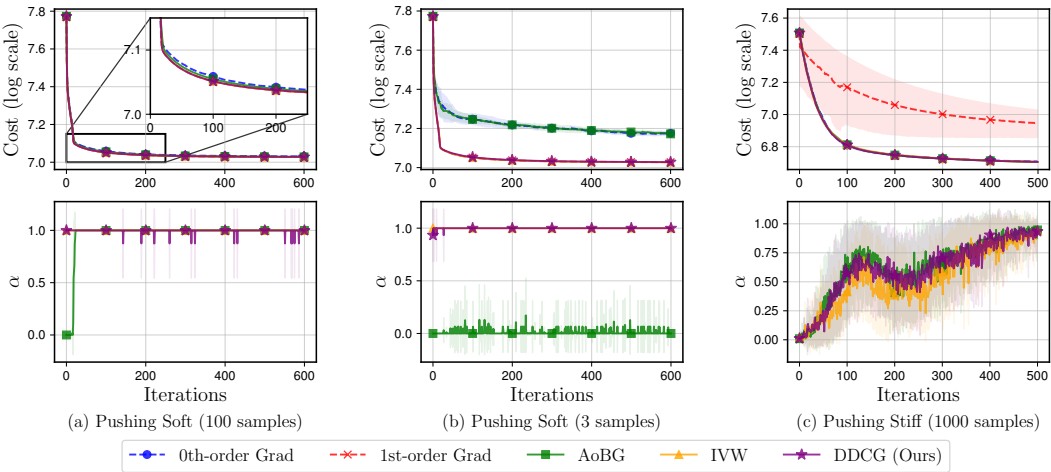

Figure 3: Pushing. Columns 1, 2: soft collisions with different samples; Column 3: stiff collisions.

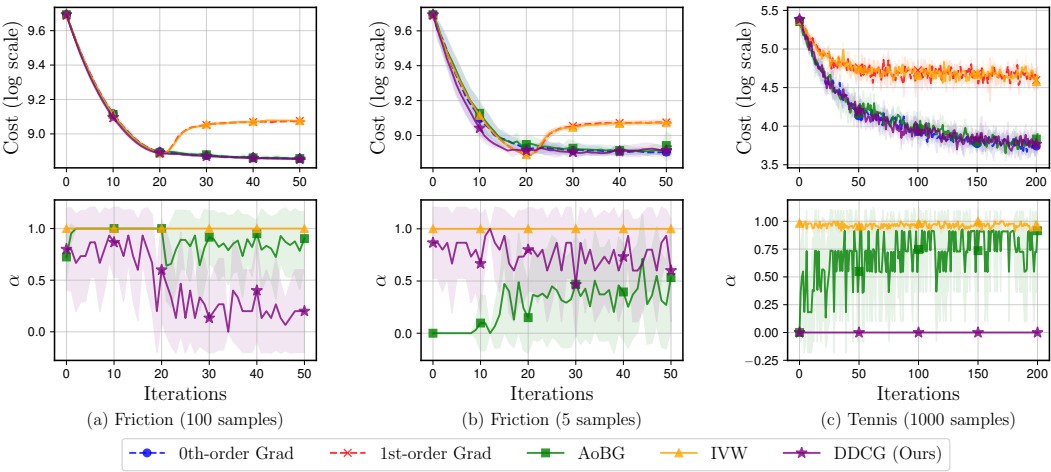

Figure 4: Columns 1, 2: Friction with different samples; Column 3: Tennis.

to the 0th-order. When reducing the sample size, as in Figure 4(b), AoBG's performance degrades unless $\gamma$ is re-tuned, while DDCG maintains robustness against small-sample noise.

### 5.2.3 POLICY OPTIMIZATION

**Tennis.** Policy optimization adjusts the parameters $\boldsymbol{\theta}$ of a state–feedback controller $\pi_{\boldsymbol{\theta}}$. The policy gradient obeys $\nabla_{\boldsymbol{\theta}} J = \nabla_{\mathbf{u}} J \, \mathbf{J}_{\pi}$, where $\mathbf{J}_{\pi} = \partial \mathbf{u} / \partial \boldsymbol{\theta}$ is the policy Jacobian. In Tennis, the agent steers a paddle to bounce an incoming ball toward a target. We optimize a linear policy of dimension $d = 21$ over horizon $H = 200$. Ball–paddle impacts create discontinuities, making the gradient unreliable in rough regions. Within DDCG (Sec. 4.1), each $\nabla f$ is instantiated as $\nabla_{\mathbf{u}} J$, and the empirical variance $\hat{v}$ in Eq. (12) is computed over samples of $\nabla_{\mathbf{u}} J$. We compare AoBG ($\gamma = 1000$) and DDCG ($c = 0.3$). Figure 4(c) shows that 1st-order and IVW stall, whereas AoBG and DDCG detect nonsmooth regimes, revert to 0th-order updates, and continue improving. AoBG and DDCG achieve identical final performance.

### 5.3 CONTINUOUS CONTROL BENCHMARKS

**Experimental Setup.** We evaluate 0th-order grad, 1st-order grad, AoBG, IVW, IVW-H, and GIPPO on MuJoCo-style tasks (*CartPole*, *Hopper*, *Ant*). Simulator and training hyperparameters follow GIPPO (Son et al., 2023). For AoBG, we set the hyperparameter $\gamma$ separately for each task based on preliminary sweeps: $\gamma = 1$ for *CartPole*, $\gamma = 10^6$ for *Ant*, and $\gamma = 10^5$ for *Hopper*. To probe whether empirical bias is the dominant issue under harder contacts, we modify only the normal

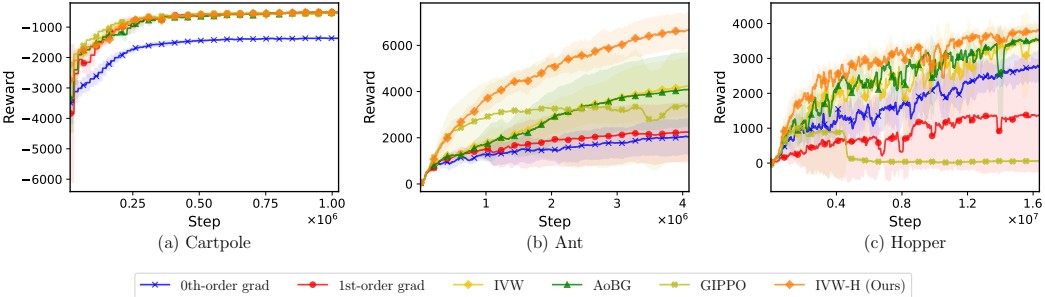

Figure 5: Episodic reward vs. environment steps on three MuJoCo-style tasks. Curves show the mean across seeds; shaded bands indicate the empirical standard error.

contact stiffness (contact_ke). Specifically, for Ant we set contact_ke $= 4.0 \times 10^5$ ($10\times$ the GIPPO value), and for Hopper we set contact_ke $= 1.0 \times 10^6$ ($50\times$). For completeness, we also report results under the original (unmodified) contact parameters in Appendix J, where both GIPPO and IVW optimize reliably and IVW-H matches or slightly improves upon IVW.

**Experimental Results.** *CartPole* (Figure 5(a)): the 0th-order baseline underperforms, while 1st-order, AoBG, IVW, IVW-H, and GIPPO reach similar final rewards. *Ant* (Figure 5(b)): IVW-H attains the best returns; AoBG performs similarly to IVW (unable to outperform it even with tuning), IVW and GIPPO are comparable and clearly above 1st-order-only and 0th-order-only, which struggle. *Hopper* (Figure 5(c)): 0th-order surpasses 1st-order-only; GIPPO fails to optimize; AoBG and IVW perform well, and IVW-H further improves upon IVW. Overall, these results indicate that variance control via stepwise IVW-H is often more critical than explicit bias detection on these benchmarks.

## 5.4 Summary of Experimental Findings

**Empirical-bias settings.** In explicitly discontinuous regimes, IVW and the 1st-order estimator exhibit clear accuracy degradation near nonsmooth events. By contrast, AoBG and DDCG avoid failures by down-weighting 1st-order information in such regions. However, inspecting AoBG's $\alpha$ trajectories indicates that its behavior is largely governed by heuristic parameter choices, with a wide operating range across tasks. In particular, AoBG requires task-specific $\gamma$ values that vary widely across our setups, with $\gamma \in [5 \times 10^{-3}, 10^8]$. DDCG maintains robustness under a unified parameter and continues to function reliably even with small sample sizes; in fact, performance was essentially unchanged for any $c \in [0.1, 0.9]$.

**Practical continuous control.** In MuJoCo-style experiments with elevated contact, the IVW-H implementation achieves strong performance and consistently improves over standard IVW. Contrary to the explicit empirical-bias settings above, these results suggest that *variance*, rather than empirical bias, is the dominant issue in these benchmarks; a practical per-step implementation such as IVW-H is sufficient to solve the problem effectively. On the MuJoCo-style tasks we consider, IVW-H matches or improves upon both IVW and the GIPPO composite-gradient baseline, without requiring any explicit empirical-bias detection.

## 6 Conclusion and Discussion

This work primarily re-examines AoBG's claims under explicit empirical-bias regimes. Reproducing the original settings, we confirm that empirical bias indeed creates failure cases for variance-based mixing, and we show that DDCG—while following the same protocol—achieves more robust behavior with a unified hyperparameter by statistically detecting nonsmooth regions and switching estimators accordingly. As a practical complement, we introduce IVW-H, a faithful per-step IVW implementation. On the MuJoCo-style benchmarks we study, IVW-H performs strongly without an explicit bias-detection scheme, indicating that in these environments variance control, rather than bias handling, is often the dominant practical concern, in contrast to the explicit empirical-bias AoBG tasks where DDCG brings clear gains. Future work will broaden the task suite and deepen diagnostics to further delineate when bias-focused mechanisms are necessary beyond such practical implementations.

AUTHOR CONTRIBUTIONS

Ku Onoda: Ran the DDCG experiments, as well as part of the experiments on the DFlex simulator tasks, discussed the interpretation of the results and wrote the paper.

Paavo Parmas: Conceptualized the idea; derived most of the equations, including the DDCG method; made the first IVW implementation in the DFlex simulator tasks; proposed the detailed implementation of IVW-H; oversaw and supervised the project; discussed the interpretation of the results; comments and editing on the writing.

Manato Yaguchi: Implemented IVW-H on the DFlex simulator tasks based on the existing IVW implementation and proposal of IVW-H by PP, and ran most of the DFlex simulator experiments. Some comments on the writing.

Yutaka Matsuo: PI of lab, funding acquisition, overall supervision in the lab.

ACKNOWLEDGMENTS

Paavo Parmas was supported by JST ACT-X, Japan, Grant Number JPMJAX23CO.

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

APPENDICES: DOES "DO DIFFERENTIABLE SIMULATORS GIVE BETTER POLICY GRADIENTS?" GIVE BETTER POLICY GRADIENTS?

## A    EXTENDED RELATED WORKS

**Policy Optimization with Differentiable Simulation.**    In this appendix, we review additional research that leverages differentiable simulators for policy optimization and clarify the positioning of our work within this broader context.

Policy Optimization via Differentiable Simulators (PODS) (Mora et al., 2021) refines policies using 1st- and 2nd-order updates derived from analytic gradients of the value function with respect to the policy actions. Analytic Policy Gradient (APG) (Freeman et al., 2021) directly computes policy gradients from simulator-provided analytic derivatives. These approaches do not explicitly consider discontinuities.

Several methods attempt to smooth the objective itself. Short-Horizon Actor-Critic (SHAC) (Xu et al., 2022) truncates trajectories to a short horizon and uses a terminal value function to smooth the objective while exploiting analytic gradients. Soft Analytic Policy Optimization (SAPO) (Xing et al., 2024) adopts a maximum-entropy RL framework and scales SHAC-style differentiable RL to deformable-body tasks, achieving superior performance over other methods on manipulation and locomotion benchmarks.

Other studies mitigate the effects of discontinuities by re-weighting analytic gradients rather than detecting them directly. Adaptive-Gradient Policy Optimization (AGPO) (Gao et al., 2024) analyzes batch-gradient variance and switches to a surrogate Q-function, ensuring convergence and robustness under non-smooth MuJoCo-style dynamics. Gradient-Informed Proximal Policy Optimization (GIPPO) (Son et al., 2023) introduces an adaptively weighted $\alpha$-policy to attenuate high-variance or biased analytic gradients, yielding consistent gains over PPO in function optimization, physics, and traffic control domains. While these methods alleviate discontinuity issues, they do not explicitly detect discontinuities.

A complementary line of work introduces explicit smoothing to handle non-smooth dynamics. Adaptive Barrier Smoothing (ABS) (Zhang et al., 2023) alleviates stiffness in complementarity-based contact models by adding barrier-smoothed objectives with an adaptive central-path parameter, jointly controlling gradient variance and bias for stable 1st-order policy gradients. By smoothing contact interactions, analytic-gradient methods such as SHAC have been applied successfully to learn physically plausible quadrupedal locomotion (Schwarke et al., 2024).

## B    INFINITE VARIANCE EXAMPLE

In this appendix, we provide a simplified example illustrating how a gradient estimator can exhibit infinite variance under a small-probability event. Suppose we have a random gradient $g(\omega)$ taking value $g_1$ with probability $p$ and 0 otherwise (with probability $1 - p$). Let $G$ be the mean of this random gradient. Then,

$$\mathbb{E}[g] = p \cdot g_1 = G \quad \Longrightarrow \quad g_1 = \frac{G}{p}. \tag{19}$$

Next, compute the second moment:

$$\mathbb{E}[g^2] \;=\; p \cdot g_1^2 + (1-p) \cdot 0^2 \;=\; p \cdot \left(\frac{G}{p}\right)^2 \;=\; \frac{G^2}{p}. \tag{20}$$

The variance $\mathbb{V}[g]$ is given by:

$$\mathbb{V}[g] = \mathbb{E}[g^2] - \left(\mathbb{E}[g]\right)^2 = \frac{G^2}{p} \;-\; G^2 = G^2 \left(\frac{1}{p} - 1\right). \tag{21}$$

As $p \to 0$, the term $\frac{1}{p}$ goes to infinity, causing $\mathbb{V}[g]$ to blow up without bound. In practice, this situation occurs when the estimator's nonzero gradients occur only in a very small region of the parameter or state space, but those gradients can be extremely large. Although the unbiasedness condition $p\,g_1 = G$ still holds, the variance is unbounded when $p$ approaches zero. This example closely parallels the situation where a Sigmoid gradients are near zero for most inputs (large $|x|$) and very large for a small range (near $x = 0$ with small temperature $T$).

## C  PROOFS

This appendix provides a step-by-step derivation of the key inequality Eq. (14) used in our proposed discontinuity-detection mechanism. We introduce a linear model for changes in gradient magnitude and then construct a quadratic model of $f(\boldsymbol{x})$. These assumptions collectively yield a condition under which IVW is expected to work well. If the condition fails, we revert to the 0th-order gradient estimator to avoid potential bias or misleading variance estimates.

FIRST INEQUALITY:

Define a linear model on the change in gradient magnitude between two points $\boldsymbol{x}$ and $\boldsymbol{y}$:

$$L \, \|\boldsymbol{x} - \boldsymbol{y}\|_2 \;\approx\; \|\nabla f(\boldsymbol{x}) - \nabla f(\boldsymbol{y})\|_2, \tag{22}$$

such that the squared difference is minimized in expectation. We thus have:

$$\mathbb{E}\Big[\frac{\partial}{\partial L}\big(L\|\boldsymbol{x} - \boldsymbol{y}\|_2 - \|\nabla f(\boldsymbol{x}) - \nabla f(\boldsymbol{y})\|_2\big)^2\Big] = 0,$$
$$\Rightarrow \mathbb{E}\big[\|\boldsymbol{x} - \boldsymbol{y}\|_2\big(L\|\boldsymbol{x} - \boldsymbol{y}\|_2 - \|\nabla f(\boldsymbol{x}) - \nabla f(\boldsymbol{y})\|_2\big)\big] = 0. \tag{23}$$

Define

$$\Delta_{\boldsymbol{x}\boldsymbol{y}} \;=\; \|\nabla f(\boldsymbol{x}) - \nabla f(\boldsymbol{y})\|_2 \;-\; L\|\boldsymbol{x} - \boldsymbol{y}\|_2. \tag{24}$$

Noting that

$$2\,\mathbb{V}[\boldsymbol{x}] \;=\; \mathbb{E}\big[\|\boldsymbol{x} - \boldsymbol{y}\|_2^2\big], \tag{25}$$

for arbitrary random variables, we can construct another equation involving the gradient differences and the above definition:

$$\begin{aligned} 2\,\mathbb{V}\big[\nabla f(\boldsymbol{x})\big] &= \mathbb{E}\big[\|\nabla f(\boldsymbol{x}) - \nabla f(\boldsymbol{y})\|_2^2\big] \\ &= \mathbb{E}\big[(L\|\boldsymbol{x} - \boldsymbol{y}\|_2 + \Delta_{xy})^2\big] \\ &= L^2\,\mathbb{E}\big[\|\boldsymbol{x} - \boldsymbol{y}\|_2^2\big] \;+\; \mathbb{E}\big[\Delta_{\boldsymbol{x}\boldsymbol{y}}^2\big] \;+\; \underbrace{2\,L\,\mathbb{E}\big[\|\boldsymbol{x} - \boldsymbol{y}\|_2 \Delta_{\boldsymbol{x}\boldsymbol{y}}\big]}_{=0 \text{ from Eq. (23)}}. \end{aligned} \tag{26}$$

Using Eq. (25) again, and noting that $\mathbb{E}\big[\Delta_{\boldsymbol{x}\boldsymbol{y}}^2\big] \geq 0$, we obtain

$$\begin{aligned} 2\,\mathbb{V}\big[\nabla f(\boldsymbol{x})\big] &\geq L^2\,\mathbb{E}\big[\|\boldsymbol{x} - \boldsymbol{y}\|_2^2\big] \\ \Rightarrow L^2 &\leq \frac{\mathbb{V}[\nabla f(\boldsymbol{x})]}{\mathbb{V}[\boldsymbol{x}]}. \end{aligned} \tag{27}$$

Furthermore, using $\mathbb{V}[\boldsymbol{x}] = D\sigma^2$, we get

$$L^2 \leq \frac{\mathbb{V}[\nabla f(\boldsymbol{x})]}{D\sigma^2}, \tag{28}$$

where $\sigma^2$ is the variance of $\boldsymbol{x}$, and $D$ is the dimension.

SECOND INEQUALITY:

Using the same quantity $L$, we will construct another inequality by making a quadratic approximation of $f(\boldsymbol{x})$. Specifically, we define a quadratic function with curvature $L$, given by

$$h(\boldsymbol{x}) = \mathbb{E}\left[f(\boldsymbol{x})\right] + \overline{\nabla f}^T(\boldsymbol{x} - \boldsymbol{\mu}) + \frac{1}{2}L\,\|\boldsymbol{x} - \boldsymbol{\mu}\|_2^2, \tag{29}$$

where $\overline{\nabla f} = \mathbb{E}\left[\nabla f(\boldsymbol{x})\right]$. We also define $\Delta f(\boldsymbol{x}) := f(\boldsymbol{x}) - h(\boldsymbol{x})$. Then, we have the equation

$$\begin{aligned} \mathbb{V}\left[f(\boldsymbol{x})\right] &= \mathbb{V}\left[h(\boldsymbol{x}) + \Delta f(\boldsymbol{x})\right] \\ &= \mathbb{V}\left[h(x)\right] + \underbrace{\mathbb{V}\left[\Delta f(\boldsymbol{x})\right] + 2\mathrm{cov}(\Delta f(\boldsymbol{x}), h(\boldsymbol{x}))}_{:=\sigma_\Delta^2} \\ &= \mathbb{V}\left[\overline{\nabla f}^T \boldsymbol{x}\right] + \mathbb{V}\left[\frac{1}{2}L\,\|\boldsymbol{x} - \boldsymbol{\mu}\|_2^2\right] + \underbrace{\mathrm{cov}\left(\overline{\nabla f}^T(\boldsymbol{x} - \boldsymbol{\mu}), \frac{1}{2}L\,\|\boldsymbol{x} - \boldsymbol{\mu}\|_2^2\right)}_{=0 \quad \text{Covariance between odd and even.}} + \sigma_\Delta^2. \end{aligned} \tag{30}$$

Now we make another assumption $\sigma_\Delta^2 < c\mathbb{V}[f(\boldsymbol{x})]$, where $c \in [0, 1]$. Then we have the inequality

$$\mathbb{V}[f(\boldsymbol{x})] \leq \mathbb{V}\left[\overline{\nabla f}^T \boldsymbol{x}\right] + \mathbb{V}\left[\frac{1}{2}L\|\boldsymbol{x} - \boldsymbol{\mu}\|_2^2\right] + c\mathbb{V}[f(\boldsymbol{x})]$$

$$\Rightarrow (1 - c)\mathbb{V}[f(\boldsymbol{x})] \leq \mathbb{V}\left[\overline{\nabla f}^T \boldsymbol{x}\right] + \mathbb{V}\left[\frac{1}{2}L\|\boldsymbol{x} - \boldsymbol{\mu}\|_2^2\right]$$

$$\Rightarrow \mathbb{V}\left[\frac{1}{2}L\|\boldsymbol{x} - \boldsymbol{\mu}\|_2^2\right] \geq (1 - c)\mathbb{V}[f(\boldsymbol{x})] - \mathbb{V}\left[\overline{\nabla f}^T \boldsymbol{x}\right]$$

$$\Rightarrow \frac{1}{4}L^2 \underbrace{\mathbb{V}\left[\|\boldsymbol{x} - \boldsymbol{\mu}\|_2^2\right]}_{\text{Gaussian distribution Eq. (34)}} \geq (1 - c)\mathbb{V}[f(\boldsymbol{x})] - \mathbb{V}\left[\overline{\nabla f}^T \boldsymbol{x}\right] \tag{31}$$

$$\Rightarrow \frac{1}{4}L^2(2D\sigma^4) \geq (1 - c)\mathbb{V}[f(\boldsymbol{x})] - \mathbb{V}\left[\overline{\nabla f}^T \boldsymbol{x}\right]$$

$$\Rightarrow L^2 \geq \frac{2(1 - c)\mathbb{V}[f(\boldsymbol{x})] - 2\mathbb{V}\left[\overline{\nabla f}^T \boldsymbol{x}\right]}{D\sigma^4}$$

$$\Rightarrow L^2 \geq \frac{2(1 - c)\mathbb{V}[f(\boldsymbol{x})] - 2\sigma^2\left\|\overline{\nabla f}\right\|^2}{D\sigma^4}$$

Combining with Eq. (27), we deduce:

$$\sigma^2 \mathbb{V}[\nabla f(\boldsymbol{x})] \geq 2(1 - c)\mathbb{V}[f(\boldsymbol{x})] - 2\sigma^2\|\overline{\nabla f}\|^2. \tag{32}$$

We then replace $\mathbb{V}[\nabla f(\boldsymbol{x})]$ with its empirical estimator $\hat{\boldsymbol{v}}$ and incorporate the allowed estimation error $\varepsilon_v$:

$$\hat{\boldsymbol{v}} + \varepsilon_v \overset{?}{\geq} \frac{2(1 - c)\mathbb{V}[f(\boldsymbol{x})]}{\sigma^2} - 2\|\overline{\nabla f}\|^2, \tag{33}$$

which is the same as Eq. (14) in the main text.

**Note on $\|\boldsymbol{x} - \boldsymbol{\mu}\|^2$ and Gaussian assumption.** Recall that

$$\mathbb{V}\left[\|\boldsymbol{x} - \boldsymbol{\mu}\|_2^2\right] = \mathbb{E}\left[\|\boldsymbol{x} - \boldsymbol{\mu}\|_2^4\right] - \left(\mathbb{E}\left[\|\boldsymbol{x} - \boldsymbol{\mu}\|_2^2\right]\right)^2. \tag{34}$$

For a Gaussian distribution, one can derive explicitly that

$\mathbb{E}\left[\|\boldsymbol{x} - \boldsymbol{\mu}\|_2^4\right] = 3\sigma^4$, and hence $\mathbb{V}\left[\|\boldsymbol{x} - \boldsymbol{\mu}\|_2^2\right] = 3\sigma^4 - \sigma^4 = 2\sigma^4$. Note that in Eq. (31), we used this particular result for Gaussian distributions. If a different sampling distribution is used, we would need to re-derive these statistical quantities or estimate them empirically from samples.

## D VARIANCE OF THE AoBG VS. DDCG TEST STATISTICS

**Motivation.** In the discontinuity detection test,

- **DDCG** uses the scaled empirical variance;
- **AoBG** forms a confidence interval for the mean gradient via the score-function statistic.

The reliability of either test is controlled by the sampling variance of its statistic. We therefore compare their **coefficients of variation**

$$\mathrm{CoV}(X) = \sqrt{\mathbb{V}\left[X\right]}/\mathbb{E}\left[X\right].$$

Our goal is to show

$$\boxed{\mathrm{CoV}_{\mathrm{AoBG}} = \Theta(d)\,\mathrm{CoV}_{\mathrm{DDCG}},}$$

meaning that the AoBG statistic is $\mathcal{O}(d)$ times noisier.

**Toy set-up.** Sample a $d$-dimensional vector $x$ from the isotropic Gaussian $\mathcal{N}(\mathbf{0}, \sigma^2 I_d)$ and evaluate the linear reward $f(x) = \sum_{j=1}^{d} x_j$. AoBG measures bias via the score-function term $\nabla_\mu \log p(x)\,f(x)$, while DDCG measures discontinuity via the (scaled) variance of $f$.

**Score function term.** Because $\log p(x) = -\|x - \mu\|^2/(2\sigma^2) + \mathrm{const}$ for a Gaussian,

$$\frac{\partial}{\partial \mu} \log p(x) = \frac{x - \mu}{\sigma^2} \xrightarrow{\mu = \mathbf{0}} \frac{x}{\sigma^2}. \tag{35}$$

Hence AoBG's per-sample statistic is

$$g(x) = \frac{f(x)\,x}{\sigma^2} = \frac{\left(\sum_j x_j\right) x}{\sigma^2}, \qquad \mathbb{E}\left[g\right] = \mathbf{1}. \tag{36}$$

**Step-by-step derivation of $\mathbb{V}\left[g\right]$.** Let $S = \sum_j x_j$. Then

$$\|g(x)\|^2 = \frac{S^2 \sum_i x_i^2}{\sigma^4} \implies \mathbb{E}\left[\|g\|^2\right] = \frac{\mathbb{E}\left[S^2 \sum_i x_i^2\right]}{\sigma^4}. \tag{37}$$

Expanding yields

$$\mathbb{E}\left[S^2 \sum_i x_i^2\right] = \sum_i \mathbb{E}\left[x_i^4\right] + \sum_{i \neq k} \mathbb{E}\left[x_i^2 x_k^2\right] \quad \text{(cross terms with odd powers vanish).} \tag{38}$$

For a univariate standard normal $z$, $\mathbb{E}\left[z^4\right] = 3\sigma^4$ and $\mathbb{E}\left[z_1^2 z_2^2\right] = \sigma^4$ when $z_1, z_2$ are independent. Hence

$$\mathbb{E}\left[\|g\|^2\right] = \frac{d \cdot 3\sigma^4 + d(d-1)\sigma^4}{\sigma^4} = d(d+2). \tag{39}$$

Therefore

$$\mathbb{V}\left[g\right] = \mathbb{E}\left[\|g\|^2\right] - \|\mathbb{E}\left[g\right]\|^2 = d(d+2) - d^2 = d(d+1) = \Theta(d^2). \tag{40}$$

**DDCG statistic.** Define $Z = \frac{\hat{\mathbb{V}}[f(x)]}{\sigma^2} = f(x)^2/\sigma^2$. Because $f(x) \sim \mathcal{N}\left(0, d\sigma^2\right)$,

$$\mathbb{E}\left[Z\right] = d, \qquad \mathbb{V}\left[Z\right] = 2d^2. \tag{41}$$

For a batch of size $n$ the statistic used by DDCG is the sample mean

$$\hat{v} = \frac{1}{n} \sum_{k=1}^{n} Z_k. \tag{42}$$

Its sampling variance is therefore

$$\mathbb{V}\left[\hat{v}\right] = \frac{\mathbb{V}\left[Z\right]}{n} = \frac{2d^2}{n}. \tag{43}$$

**Relative precision (coefficient of variation).** For any statistic $X$ we define

$$\text{CoV}(X) = \sqrt{\mathbb{V}\left[X\right]}\big/\mathbb{E}\left[X\right]. \tag{44}$$

Hence

$$\text{CoV}_{\text{DDCG}} = \frac{\sqrt{2d^2/n}}{d} = \sqrt{\frac{2}{n\,d}}, \qquad \text{CoV}_{\text{AoBG}} = \frac{\sqrt{d+1}/\sqrt{n}}{1} \approx \sqrt{\frac{d}{n}}, \tag{45}$$

and their ratio scales as

$$\frac{\text{CoV}_{\text{AoBG}}}{\text{CoV}_{\text{DDCG}}} = \frac{\sqrt{d/n}}{\sqrt{2/(n\,d)}} = \frac{d}{\sqrt{2}} = \Theta(d). \tag{46}$$

Thus AoBG's statistic is $\mathcal{O}(d)$ times noisier than DDCG's, demonstrating DDCG's advantage in high-dimensional settings.

**Monte-Carlo confirmation.** CoV quantifies the relative estimation error: it is the standard deviation of the statistic divided by its mean. We ran $m = 10000$ independent batches of size $n = 1000$ with $\sigma = 1$; Table 1 reports the empirical CoVs. The ratio $\text{CoV}_{\text{AoBG}}/\text{CoV}_{\text{DDCG}}$ decays approximately as $d$, confirming the theoretical gap.

Table 1: Precision of the two test statistics ($n{=}1000$, $m{=}10000$, $\sigma{=}1$).

| $d$ | $\text{CoV}_{\text{DDCG}}$ | $\text{CoV}_{\text{AoBG}}$ | ratio | ratio $\times \sqrt{2}$ |
|---|---|---|---|---|
| 1 | 4.49e-2 | 4.47e-2 | 1.00 | 1.41 |
| 16 | 1.11e-2 | 1.30e-1 | 11.7 | 16.6 |
| 64 | 5.56e-3 | 2.55e-1 | 45.5 | 64.4 |
| 128 | 3.90e-3 | 3.59e-1 | 92.2 | 130 |

# E    PSEUDOCODE FOR IVW-H

We implement a practical composite update that combines 0th- and 1st-order policy gradients at the *step* and *action-dimension* level. The procedure is summarized in Alg. 1.

---

**Algorithm 1** IVW-H Policy Update (stepwise IVW)

---

**Require:** Horizon $H$, actors $N$, action dim. $A$; policy $\pi_{\boldsymbol{\theta}}$ (Gaussian: $\boldsymbol{\mu}, \boldsymbol{\sigma}$); target critic $\hat{V}$; advantages $\mathbf{A}_t$ via GAE; mask $\texttt{grad\_start} \in \{0,1\}^{H \times N}$ for first-terms; optional pairwise noise/initial-state sharing.

1: Define $s_{t,n} := \texttt{grad\_start}[t,n]$ and $M := \sum_{t=0}^{H-1} \sum_{n=1}^{N} s_{t,n}$ ▷ number of trajectories (episode starts) in the batch

2: **Rollout.** For $t = 0, \ldots, H-1$: compute $(\boldsymbol{\mu}_t, \boldsymbol{\sigma}_t) = \pi_{\boldsymbol{\theta}}(\mathbf{s}_t)$, sample $\boldsymbol{\epsilon}_t \sim \mathcal{N}(0, I)$, act $\mathbf{a}_t = \tanh(\boldsymbol{\mu}_t + \boldsymbol{\sigma}_t \odot \boldsymbol{\epsilon}_t)$, step envs, cache $\{\mathbf{s}_t, \mathbf{a}_t, \boldsymbol{\mu}_t, \boldsymbol{\sigma}_t\}$, and mark $\texttt{grad\_start}$ at episode starts.

3: **Advantages.** Using $\{r_t, \hat{V}\}$, compute GAE $\mathbf{A}_t$; define the first-term sum over starts.

4: **Losses exposing $g_1$ and $g_0$.**

  - *RP/1st-order loss:* $\mathcal{L}_{\mathrm{rp}} \leftarrow -\dfrac{1}{M} \displaystyle\sum_{t,n:\, s_{t,n}=1} \mathbf{A}_{t,n}.$     *(normalize by number of trajectories $M$)*

  - *LR/0th-order loss:* $\mathcal{L}_{\mathrm{lr}} \leftarrow \mathrm{mean}_{t,n}(\tilde{\mathbf{A}}_{t,n} \odot \mathrm{neglogp}_{t,n})$ with optional normalization of $\tilde{\mathbf{A}}$. *(batch mean)*

5: **Parameter-level gradients (per step and per dimension).**

  Backprop $\mathcal{L}_{\mathrm{rp}} \Rightarrow \hat{\boldsymbol{g}}^1_{t,n,a,\phi} \equiv \partial \mathcal{L}_{\mathrm{rp}} / \partial \phi_{t,n,a}$,     Backprop $\mathcal{L}_{\mathrm{lr}} \Rightarrow \hat{\boldsymbol{g}}^0_{t,n,a,\phi} \equiv \partial \mathcal{L}_{\mathrm{lr}} / \partial \phi_{t,n,a}$,

  where $\phi \in \{\mu, \sigma\}$ and $(t, n, a)$ index time, actor, and action dim.

6: **Stepwise variance across actors.** $\hat{\boldsymbol{v}}^0_{t,a,\phi} = \hat{\mathbb{V}}_n\big[\hat{\boldsymbol{g}}^0_{t,n,a,\phi}\big], \quad \hat{\boldsymbol{v}}^1_{t,a,\phi} = \hat{\mathbb{V}}_n\big[\hat{\boldsymbol{g}}^1_{t,n,a,\phi}\big].$

7: **IVW-H fusion (per step, per dimension).**

$$\hat{\boldsymbol{\alpha}}_{t,a,\phi} = \frac{\hat{\boldsymbol{v}}^0_{t,a,\phi}}{\hat{\boldsymbol{v}}^0_{t,a,\phi} + \hat{\boldsymbol{v}}^1_{t,a,\phi}}, \quad \boldsymbol{G}_{t,n,a,\phi} = \hat{\boldsymbol{\alpha}}_{t,a,\phi}\, \hat{\boldsymbol{g}}^1_{t,n,a,\phi} + \big(1 - \hat{\boldsymbol{\alpha}}_{t,a,\phi}\big)\, \hat{\boldsymbol{g}}^0_{t,n,a,\phi},$$

  with $\hat{\boldsymbol{\alpha}}_{t,a,\phi} \leftarrow 0$ wherever the DDCG gate suppresses $g_1$.

8: **Push to policy weights.** Treat $\{\boldsymbol{G}_{t,n,a,\phi}\}$ as the target gradient on distribution parameters and perform a vector–Jacobian product through $\pi_{\boldsymbol{\theta}}$ to obtain $\nabla_{\boldsymbol{\theta}} \mathcal{L}$. Apply clipping if needed and update $\boldsymbol{\theta}$ with Adam.

9: **Critic.** Fit $\hat{V}$ by MSE to targets $\mathbf{A}_t + \hat{V}(\mathbf{s}_t)$.

---

# F    FUNCTION OPTIMIZATION TASKS

We measure the gradient estimation error on simple functions, revealing how each method adapts $\alpha$ under varying degrees of discontinuity and sample sizes.

**Experimental Setup.** We evaluate two functions (Sigmoid and Quadratic). For the sigmoid function, we vary the temperature $T$, where smaller $T$ yields near-discontinuities. For both, we also vary the sample size $N$ to evaluate how each method performs with limited samples. For AoBG, the parameters $\gamma$ is tuned separately for each function so that the methods perform well when the sample size is sufficiently large (around 100). Specifically, for the Sigmoid, we set $\gamma = 0.1$, and for the Quadratic, we set $\gamma = 1.4$. In contrast, DDCG uses the same settings ($c = 0.3$) across all toy tasks. For detailed parameter settings, see Appendix K.

**Findings.** In Figure 6(a), as the Sigmoid transitions become sharper (i.e., for smaller $T$), IVW starts to over-rely on 1st-order gradients and becomes biased. Both AoBG and DDCG detect these discontinuities and shift more weight to 0th-order, reducing error. However, as shown in Figure 6(b) and (c), AoBG tends to assign conservative weights to the 0th-order component when sample sizes are small, causing the weighting parameter $\alpha$ to drop. This behavior arises from its sensitivity to the hyperparameter $\gamma$; without re-tuning, AoBG may underutilize useful 1st-order gradients, missing potential performance gains. DDCG, in contrast, maintains robust performance across both smooth and near-discontinuous regimes, achieving comparable or better error reduction with a fixed parameter setting.

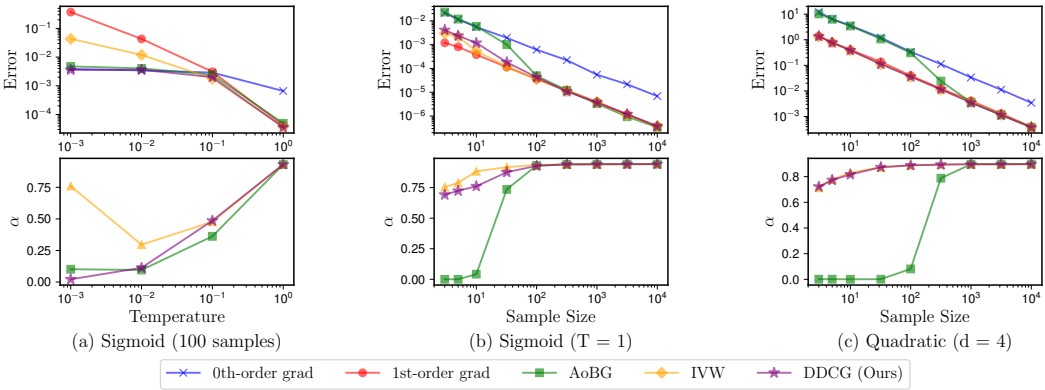

(a) Sigmoid (100 samples)    (b) Sigmoid ($T = 1$)    (c) Quadratic ($d = 4$)

0th-order grad    1st-order grad    AoBG    IVW    DDCG (Ours)

Figure 6: Performance analysis for Sigmoid (Columns 1, 2) and Quadratic (Column 3) functions under varying temperatures and sample sizes. Top row: estimation errors (log scale) between true and estimated gradients for each method. Bottom row: weighting parameter $\alpha$ for each method, showing selection between 0th- and 1st-order gradients.

# G    ADDITIONAL EXPERIMENTS

In this appendix, we provide more detailed results from the landscape analysis in Section 5.2.1 for the Ball with Wall task, including variance and bias components of the gradient estimation error. We also present analogous results for the Momentum Transfer task, which could not be shown in the main text.

## G.1    BALL WITH WALL TASK

As shown in Figure 7, near discontinuities, the 1st-order gradient estimator exhibits a large bias that dominates the overall error. When the sample size is sufficiently large ($N = 1000$), variance does not pose a significant problem. However, with fewer samples, the 0th-order estimator tends to have higher variance, making it crucial to switch adaptively between 1st- and 0th-order estimates. DDCG achieves this by emphasizing the 1st-order gradient in smooth regions to reduce variance while switching to 0th-order near discontinuities to avoid bias, thus maintaining low error across the entire landscape.

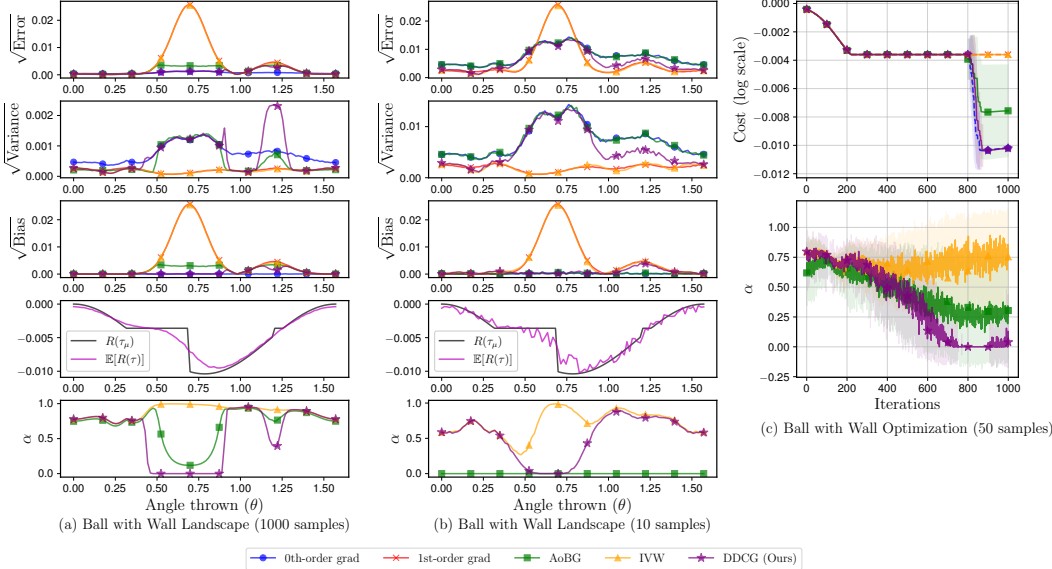

Figure 7: Ball with Wall task. Columns 1 and 2: The first to third rows show the square root of estimation errors, variance, and bias, respectively (scaled to match the previous study). The fourth row shows the cost function, and the bottom row shows $\alpha$ selection. Column 3: Both the optimization cost and $\alpha$ selection are shown.

## G.2 MOMENTUM TRANSFER TASK

Figure 8 shows similar results for the Momentum Transfer task. In terms of cost minimization, just as in Ball with Wall, the 1st-order gradient and IVW struggle with discontinuities, whereas the other methods successfully circumvent them.

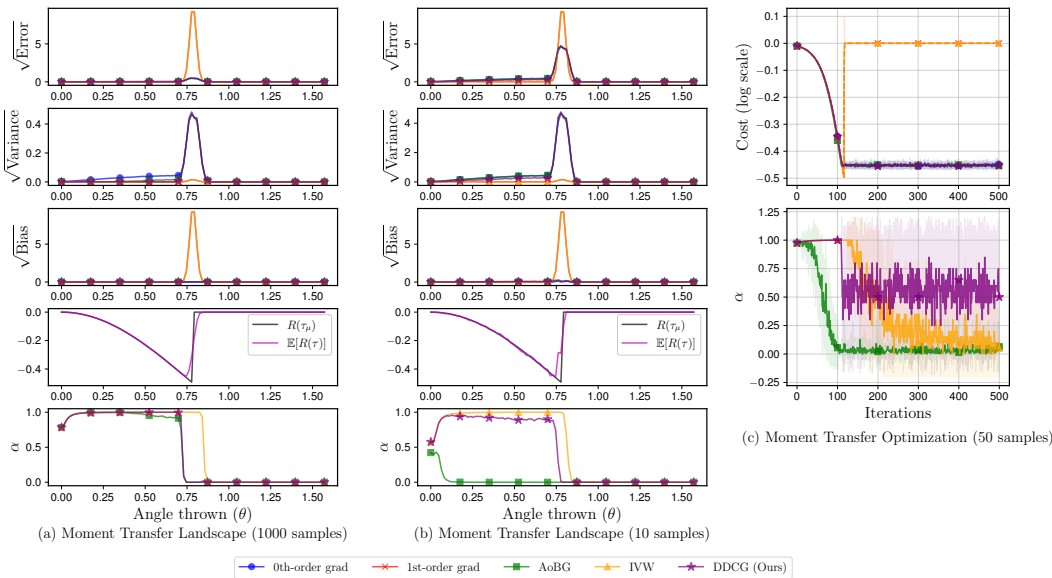

Figure 8: Momentum Transfer task. Columns 1 and 2: The first to third rows show the square root of estimation errors, variance, and bias, respectively (scaled to match the previous study). The fourth row shows the cost function, and the bottom row shows $\alpha$ selection. Column 3: Both the optimization cost and $\alpha$ selection are shown.

## H    SENSITIVITY ANALYSIS ON THE PARAMETER $c$ IN DDCG

In this section, we conduct a sensitivity analysis on the parameter $c$ in our proposed DDCG method to investigate how varying $c$ affects the detection of discontinuities. We also clarify why $c = 0.3$ was chosen in this work.

### H.1    BALL WITH WALL LANDSCAPE

Figure 9 visualizes the Ball with Wall task landscape while varying $c$ from 0 to 1. Recall that $c = 1$ means our test condition is always satisfied, so the method consistently applies IVW, disabling discontinuity detection. Conversely, $c = 0$ imposes a strong smoothness assumption, frequently falling back to the 0th-order estimator and leading to more conservative updates. For any $c \neq 1$, the largest cost change near $\theta = 0.7$ is reliably detected. However, detecting a milder discontinuity around $\theta = 1.2$ depends on $c$. Balancing these, we set $c = 0.3$ to avoid being overly conservative or too permissive, successfully detecting both major and moderate discontinuities.

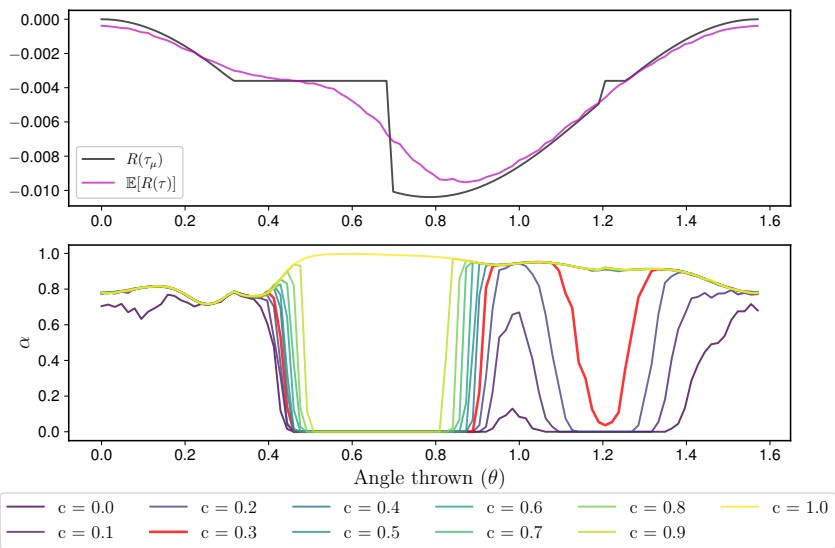

Figure 9: Sensitivity analysis on $c$ in the Ball with Wall task. The x-axis represents $\theta$, and different values of $c$ control the degree of discontinuity detection. Larger values of $c$ are less conservative, while smaller values lead to more frequent selection of 0th-order gradients.

### H.2    SIGMOID FUNCTION

Figure 10 presents a similar sensitivity analysis for the Sigmoid function, where we adjust its temperature parameter $T$. Smaller $T$ values yield sharper transitions (stronger discontinuities). For $c = 0$, DDCG assumes stronger smoothness and thus tends to remain conservative even in the $T = 1$ regime, resulting in larger estimation errors compared to larger $c$ values. On the other hand, when $c$ is close to 1, the method still detects strong discontinuities adequately, though it becomes less conservative in potentially nonsmooth areas.

### H.3    OPTIMIZATION PROBLEMS

We report a sensitivity sweep of the sole hyperparameter $c$ on the *optimization* problems: **Pushing-Soft**, **Pushing-Stiff**, **Friction**, and **Tennis**. Across these tasks, performance is *robust* for a wide range $c \in [0.1, 0.9]$—the optimizer converges reliably and at similar rates. At the extremes, $c \approx 0$ can be overly conservative on smoother tasks (e.g., *Pushing-Soft*), frequently falling back to 0th-order updates even when first-order gradients are reliable, whereas $c \approx 1$ becomes very permissive and may under-detect mild nonsmoothness on strongly non-smooth tasks (e.g., *Tennis*, *Friction*). Hence, choosing $c = 0.3$ is *representative* rather than critical.

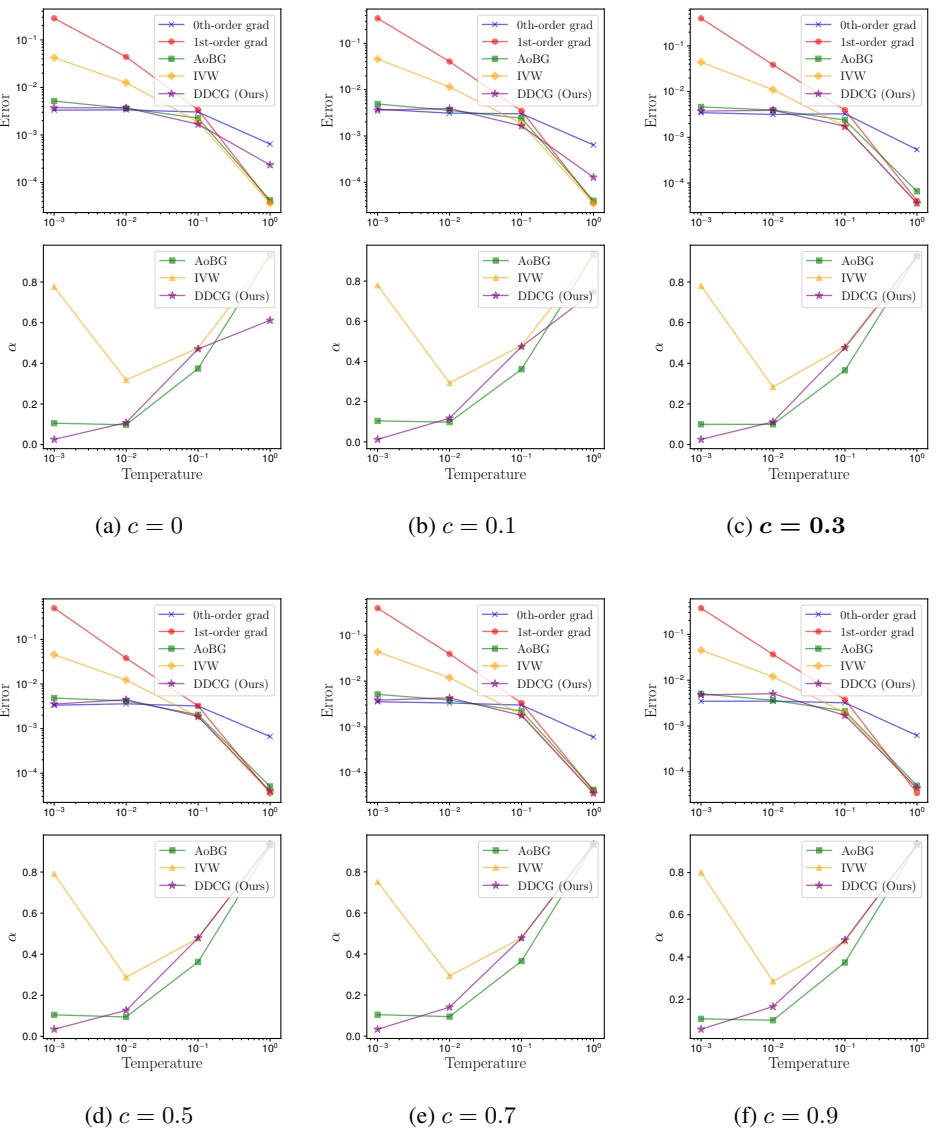

Figure 10: Sensitivity analysis on $c$ for gradient estimation error (log scale) and $\alpha$ selection in the Sigmoid function. The x-axis represents different values of the temperature parameter $T$, where smaller $T$ indicates stronger discontinuities. Lower $c$ values lead to conservative choices, while higher values make the method more permissive in discontinuity detection.

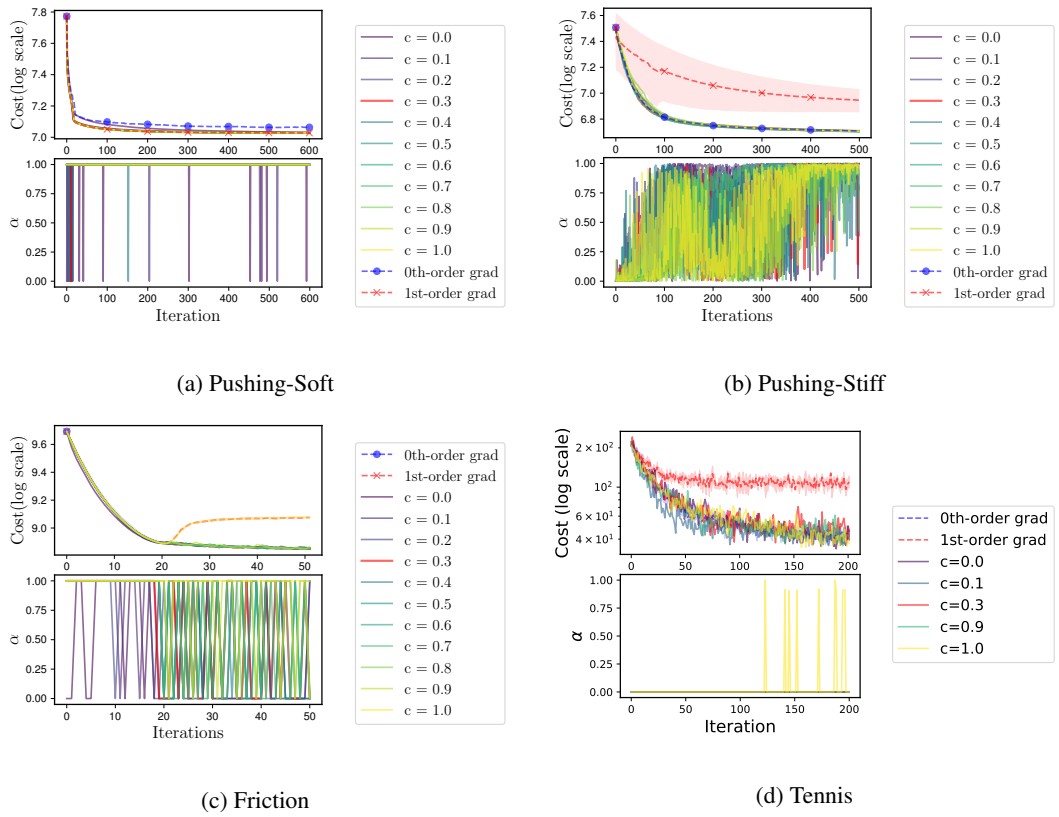

(a) Pushing-Soft

(b) Pushing-Stiff

(c) Friction

(d) Tennis

Figure 11: **Sensitivity of $c$ on optimization tasks.** Each panel shows optimization progress (e.g., objective vs. iterations or episodes) for multiple $c$ values. Results indicate that *non-extreme $c$* values yield near-identical performance; $c=0.3$ is a convenient default rather than a crucial choice.

**Takeaway.** For all optimization problems considered, DDCG solves the tasks reliably for any *non-extreme $c$* in $[0.1, 0.9]$. Thus, the method does not rely on a finely tuned $c$; using $c=0.3$ is a safe and representative default.

Overall, we found $c = 0.3$ effectively balances performance in both highly discontinuous and smoothly varying scenarios; hence, we adopt it as the default setting.

# I  SENSITIVITY ANALYSIS ON THE PARAMETER $\gamma$ IN AOBG

We conduct the sensitivity analysis on the $\gamma$ parameter of the previous method AoBG. If $\gamma$ is large, AoBG will mainly use the IVW rule, if $\gamma$ is small, AoBG mainly uses 0th-order estimates. Thus, in tasks where 0th-order estimates work well, $\gamma$ should be sufficiently small, and in tasks where 1st-order estimates are better, $\gamma$ has to besufficiently large. Ball with Wall (1000 samples) requires roughly Figure 12 and Momentum Transfer (1000 samples) requires Figure 13. In the 3-sample Pushing Soft task, 0th-order methods perform poorly, and we find that $\gamma$ should be above roughly 50000 for good performance Figure 14. On the other hand, the Tennis task performs poorly when the gamma is that large, it requires roughly Figure 15. As we can see, the optimal choice of $\gamma$ varies widely between different tasks and also changes with the sample size.

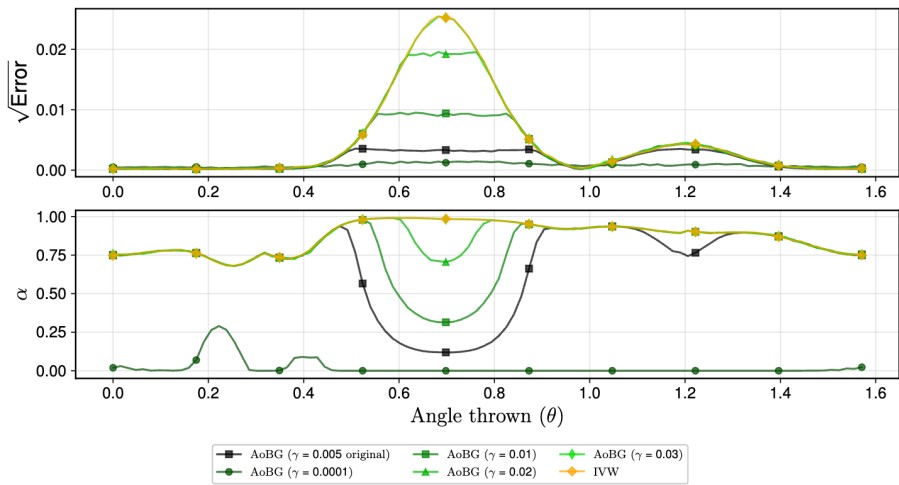

Figure 12: Sensitivity analysis on the parameter $\gamma$ for AoBG in the Ball with Wall landscape analysis (1000 samples). The figure shows the error for each input angle $\theta$ and the corresponding $\alpha$ selection.

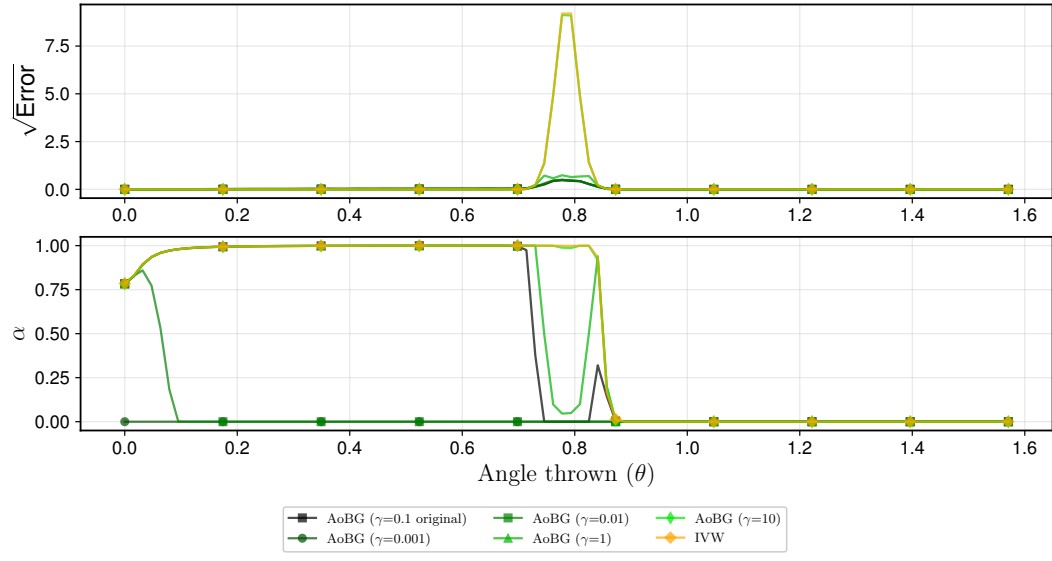

Figure 13: Sensitivity analysis on the parameter $\gamma$ for AoBG in the Momentum Transfer landscape analysis (1000 samples). The figure shows the error for each input angle $\theta$ and the corresponding $\alpha$ selection.

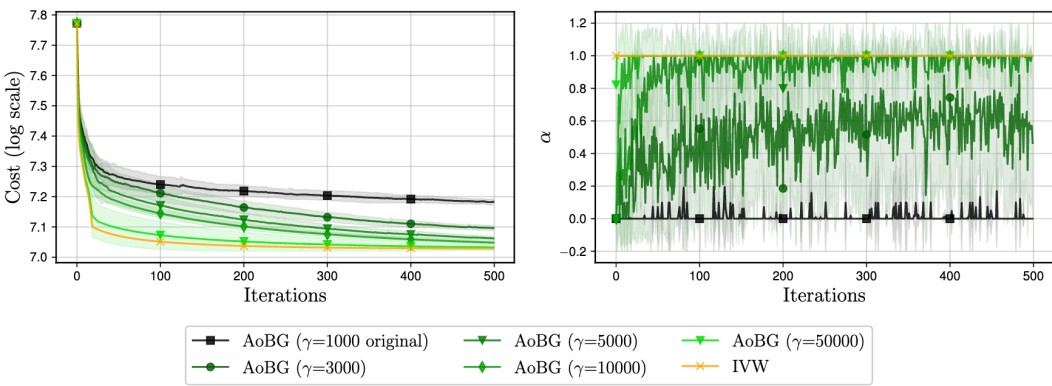

Figure 14: Sensitivity analysis on the parameter $\gamma$ for AoBG in the Pushing task with soft contact (3 samples). The figure shows the cost value evolution and the corresponding $\alpha$ selection across iterations.

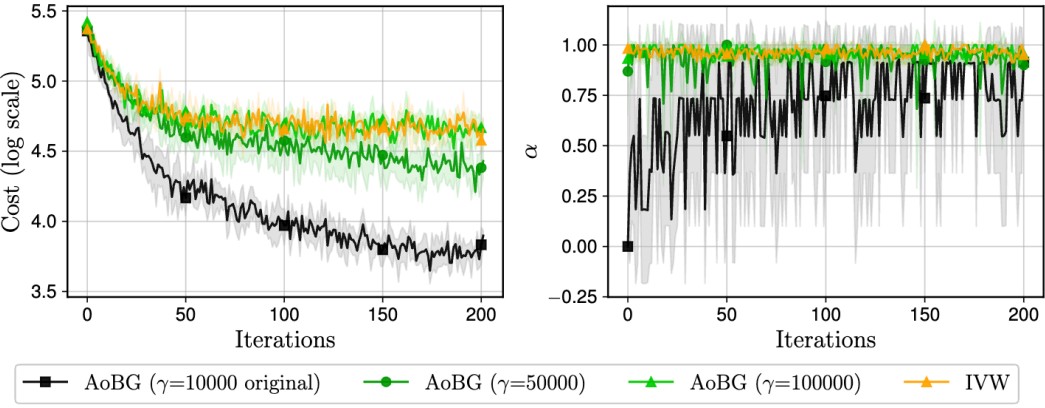

Figure 15: Sensitivity analysis on the parameter $\gamma$ for AoBG in the Tennis task. The figure shows the cost value evolution and the corresponding $\alpha$ selection across iterations.

**High-Contact MuJoCo-style Tasks.** We conducted a sensitivity analysis on $\gamma$ for the Ant and Hopper tasks with high contact setting.

Figure 16 presents the learning curves and the evolution of $\alpha$ across a wide range of $\gamma$ values ($\gamma \in \{10, \ldots, 10^6\}$). As shown in the results, while higher values of $\gamma$ generally lead to better performance, AoBG does not outperform the IVW baseline in either environment. If the performance limitation were primarily due to empirical bias, we would expect a specific range of $\gamma$ to effectively mitigate this bias and surpass the baseline. However, the fact that IVW remains competitive or superior regardless of $\gamma$ tuning suggests that empirical bias is not the dominant factor hindering performance in these specific high-contact settings.

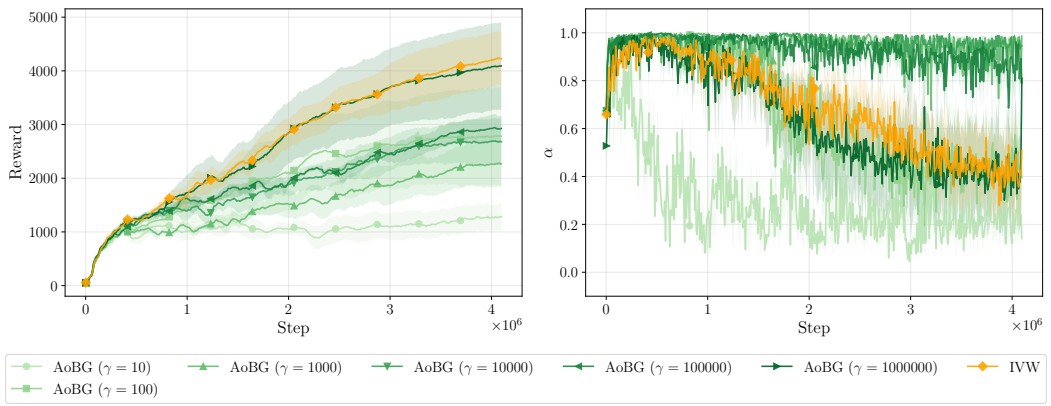

(a) Ant (High-contact scenario)

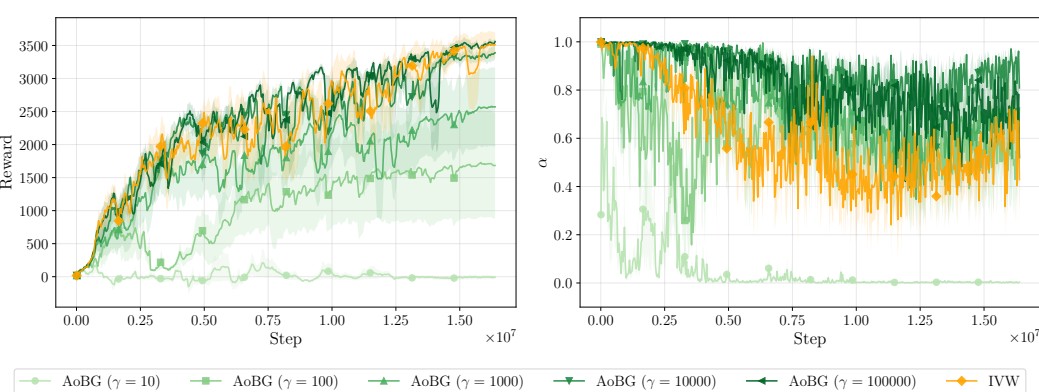

(b) Hopper (High-contact scenario)

Figure 16: **Sensitivity analysis on the parameter $\gamma$ for AoBG in high-contact environments.** We compare AoBG with varying $\gamma$ against the IVW baseline. The left plots show the learning curves (Reward), and the right plots show the evolution of the mixing coefficient $\alpha$. Notably, even with extensive tuning of $\gamma$, AoBG does not consistently outperform IVW. This suggests that the performance bottleneck in these high-contact tasks is likely not attributed to empirical bias.

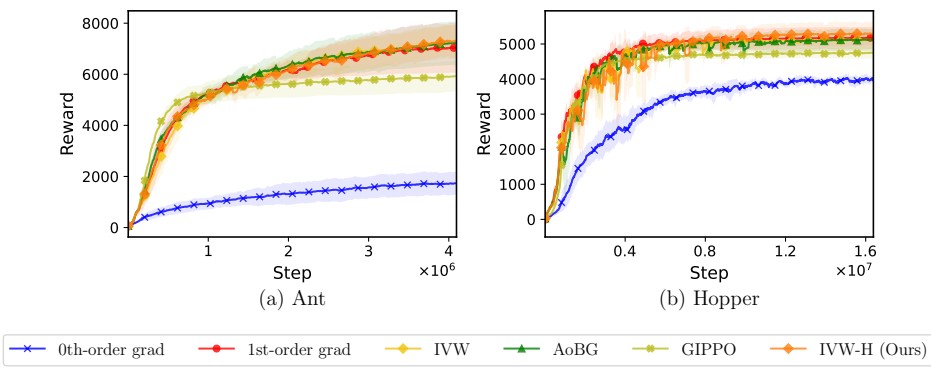

(a) Ant  (b) Hopper

Figure 17: Episodic reward vs. environment steps on *Ant* and *Hopper* with *default* contact parameters. Curves show the mean across seeds; shaded bands indicate the empirical standard error.

## J    MuJoCo-style Tasks Results with Default Contacts

Under the *default* MuJoCo-style contact settings (no change to `contact_ke`), both GIPPO and IVW optimize reliably; IVW-H matches upon IVW across *Ant* and *Hopper*, while 0th-order gradients lag behind (see Figure 17).

## K    Parameters

The following tables summarize the parameter settings used in our experiments. These parameters were chosen to ensure consistency and reproducibility across all tasks.

Table 2: Sigmoid, Quadratic parameter settings

| Parameter names | Sigmoid (Effect of Temperatures) | Sigmoid (Effect of Samples) | Quadratic (Effect of Samples) |
|---|---|---|---|
| **Common Parameters** | | | |
| Sample size $N$ | 100 | - | - |
| Standard deviation $\sigma$ | 1 | 1 | 1 |
| Trials (Seeds) | 500 | 500 | 500 |
| **AoBG** | | | |
| $\gamma$ | 0.1 | 0.1 | 1.4 |
| **DDCG** | | | |
| $c$ | 0.3 | 0.3 | 0.3 |
| Confidence level $\delta$ | 0.05 | 0.05 | 0.05 |

Table 3: Ball With Wall parameter settings

| Parameter names | Landscape (1000 samples) | Landscape (10 samples) | Optimization |
|---|---|---|---|
| **Common Parameters** | | | |
| Sample size $N$ | 1000 | 10 | 50 |
| Standard deviation $\sigma$ | 0.1 | 0.1 | 0.1 |
| Trials (Seeds) | 1000 | 1000 | 20 |
| Iterations | - | - | 1000 |
| **AoBG** | | | |
| $\gamma$ | 0.005 | 0.005 | 0.014 |
| **DDCG** | | | |
| $c$ | 0.3 | 0.3 | 0.3 |
| Confidence level $\delta$ | 0.05 | 0.05 | 0.05 |

Table 4: Momentum Transfer parameter settings

| Parameter names | Landscape (1000 samples) | Landscape (10 samples) | Optimization |
|---|---|---|---|
| **Common Parameters** | | | |
| Sample size $N$ | 1000 | 10 | 50 |
| Standard deviation $\sigma$ | 0.02 | 0.02 | 0.02 |
| Trials (Seeds) | 1000 | 1000 | 20 |
| Iterations | - | - | 5000 |
| **AoBG** | | | |
| $\gamma$ | 0.2 | 0.2 | 0.2 |
| **DDCG** | | | |
| $c$ | 0.3 | 0.3 | 0.3 |
| Confidence level $\delta$ | 0.05 | 0.05 | 0.05 |

Table 5: Pushing parameter settings

| Parameter names | Soft Collisions (100 samples) | Soft Collisions (3 samples) | Stiff Collisions |
|---|---|---|---|
| **Common Parameters** | | | |
| Sample size $N$ | 100 | 3 | 10 |
| Standard deviation $\sigma$ | 0.1 | 0.1 | 0.05 |
| Trials (Seeds) | 100 | 100 | 20 |
| Iterations | 600 | 600 | 500 |
| Spring constant $k$ | 10 | 10 | 1000 |
| **AoBG** | | | |
| $\gamma$ | 1000 | 1000 | 10000000 |
| **DDCG** | | | |
| $c$ | 0.3 | 0.3 | 0.3 |
| Confidence level $\delta$ | 0.05 | 0.05 | 0.05 |

Table 6: Friction parameter settings

| Parameter names | Trajectory (100 samples) | Trajectory (5 samples) |
|---|---|---|
| **Common Parameters** | | |
| Sample size $N$ | 100 | 5 |
| Standard deviation $\sigma$ | 0.1 | 0.1 |
| Trials (Seeds) | 15 | 15 |
| Iterations | 50 | 50 |
| **AoBG** | | |
| $\gamma$ | 30000 | 30000 |
| **DDCG** | | |
| $c$ | 0.3 | 0.3 |
| Confidence level $\delta$ | 0.05 | 0.05 |

Table 7: Tennis parameter settings

| Parameter names | Policy |
|---|---|
| **Common Parameters** | |
| Sample size $N$ | 1000 |
| Standard deviation $\sigma$ | 0.01 |
| Trials (Seeds) | 4 |
| Iterations | 200 |
| **AoBG** | |
| $\gamma$ | 1000 |
| **DDCG** | |
| $c$ | 0.3 |
| Confidence level $\delta$ | 0.05 |

