# OpenReview forum: "Does “Do Differentiable Simulators Give Better Policy Gradients?” Give Better Policy Gradients?"
_ICLR.cc/2026/Conference — ICLR 2026 Poster_

### Official Review · Reviewer_ecYo · 2025-10-28

**Soundness:** 4
**Presentation:** 4
**Contribution:** 4
**Rating:** 8
**Confidence:** 4

**Summary:**

The paper studies the problem of finding optimal trajectories using differentiable simulators.

Differentiable simulators allow us to differentiate dynamics (and rewards) w.r.t. actions, allowing efficient policy optimization. However, gradients are inaccurate in the presence of dynamic discontinuities (e.g., contact), posing a significant challenge for the use of gradient information in trajectory (or policy) optimization.

Previous work proposed mitigating the issue by combining zero-order (e.g., REINFORCE) and first-order methods (e.g., reparametrization gradients that differentiate the simulator). Zero-order gradients are unbiased, but typically suffer from high variance. First-order gradients, on the contrary, are subject to low variance, but can sporadically be subject to high variance in steep or non-differentiable regions.

In particular, the authors consider two algorithms: Inverse Variance Weighting (IVW) and Interpolation Protocol (AoBG).

IVW uses the empirical variance of the estimator to reduce variance, allocating a higher weight to the estimator with the smaller variance. This method usually allocates a higher weight to the first-order method, but still fails to remove the bias.

AoBG introduces an additional safeguard mechanism that aims to check the presence of bias by allocating more weight to the zeroth order in case the first order and the zeroth order estimates largely disagree. However, according to the authors, the hyperparameters introduced by AoBG are subject to high sensitivity and should be tuned for different environments/tasks.

Building on these intuitions, the authors propose an algorithm that also builds on IVW and, similarly to AoBG, a "filtering" that allows one to revert to the zeroth-order information in the presence of bias. The authors notice that steep dynamics not only introduce a bias but might also harm the variance estimation of the estimator (i.e., the empirical variance might not be accurate). They devise a test that checks at once that the dynamics are not discontinuous, while the empirical variance is trustworthy (eq. 14). If that is the case, they rely on IVW weighting to perform the gradient estimation; otherwise, they revert entirely to relying on the zeroth-order estimator.

The authors show that a) their algorithm is less sensitive to estimation errors due to discontinuities (Figures 2 and 3); b) Figure 4 clearly highlights the ability of the developed algorithm to switch to the zeroth order derivative when needed (unlike IVW), while AoBG can be either too conservative or too lenient. Although the error difference in the ablation studies does not seem so significant, the policy optimization experiment (Figure 5) really shows the advantage of relying on good gradient estimation. This advantage is greater in high-dimensional, contact-rich tasks like Ant and Hopper, whereas it is negligible in the smooth, low-dimensional dynamics of the pendulum.

**Strengths:**

The paper analyses an important and relevant problem in model-based policy optimization arising from inaccuracies in differentiable simulators.

The authors present the state of the art well, explain the problems, introduce their solution, and build upon it. The reading is clear and interesting.

The presented method is sound.

The empirical analysis is very well done. The authors present three studies: 1) they check how a discontinuity affects the gradient estimators; 2) they check both pure trajectory optimization and policy optimization, showing gradient estimation errors, the weights of the estimators, and the total cost; and 3) the learning curve in a pure policy optimization scenario in three MuJoCo tasks. A hyperparameter sensitivity analysis is provided in the Appendix.

The results agree with the intuition built into the method.

**Weaknesses:**

This paper has a few weaknesses:

1) Equation 14, which is the core of the method, is not really explained in the main paper. The authors should have devoted some effort in building an intuition for it, rather than relegating the proof to the appendix. I am aware that, in estimating the gradient of a function, the concavity (or convexity) has an impact on its variance, thus I think that building an intuition on eq. 14 should not be too hard.

2) The acronym AoBG has never been introduced.

3) Equations 4 and 6 seem to be one-sample estimates, but later in the paper, it becomes clear that they are utilizing n-samples.

4) The notation in equation 3--6 is a bit too compact, hiding details and dependencies between variables.

5) the number of seeds used is not explicitly stated in the main text (unless I have missed it). The tales in Appendix K report number of trials (which is typically high). I am unsure whether the trial correspond to the number of seeds.

**Questions:**

Questions
--------------

I am unsure why the proposed method checks only the empirical variance of the first-order estimate. Typically, zeroth-order estimators also suffer from high variance, which can make the empirical variance less precise. Could the authors clarify that?

Suggestions
----------------

A technical note: the baseline subtraction can make REINFORCE biased if it is estimated with the same samples. Look at this example, where b is the average reward (typical baseline subtraction):

$$\hat{g}^{(n)} = n^{-1}\sum_{i=1}^n \nabla \log p(\tau_i) \left(r(\tau_i) - \hat{b}^{(n)}\right)$$ with
$$\hat{b}^{(n)} = n^{-1} \sum_{i=1}^n r(\tau_i) $$

For $n=1$, $\hat{g}^{(n)} =0$, clearly show that, unless the ground truth is always zero, this gradient estimator is biased. Of course, this issue is avoided when one uses two independent sets of samples to estimate the gradient and the baseline.

I recommend adding this useful information to the text and explicitly writing how you compute the baseline subtraction for the 0th order.

Section 4.2 could have a more meaningful title.

---

> ### Author Response · Authors · 2025-11-21
>
> Thank you for the review.
>
> >Equation 14, which is the core of the method, is not really explained in the main paper. The authors should have devoted some effort in building an intuition for it, rather than relegating the proof to the appendix. I am aware that, in estimating the gradient of a function, the concavity (or convexity) has an impact on its variance, thus I think that building an intuition on eq. 14 should not be too hard.
>
> To address the comment, we have added a paragraph immediately following Equation 14 in Section 4.1 to provide the requested intuition. Specifically, we explain that the inequality compares the empirical gradient variance against a theoretical lower bound derived from a local quadratic approximation. If the observed variance is suspiciously low compared to this smooth baseline (violating the inequality), it signals a discontinuity or 'empirical bias' where finite samples have missed a sharp peak, thus triggering the fallback to the 0th-order estimator.
>
> >The acronym AoBG has never been introduced.
>
> We added the introduction.
>
> >Equations 4 and 6 seem to be one-sample estimates, but later in the paper, it becomes clear that they are utilizing n-samples.
>
> We have added a 'Batch Estimation' paragraph in Section 3 explicitly stating that while the equations define single-sample estimators, our practical implementation uses batch statistics.
>
> >The notation in equation 3--6 is a bit too compact, hiding details and dependencies between variables.
>
> We have revised Equations 3-6 to explicitly show the dependency on $\theta$ and the underlying probability distributions, improving mathematical clarity.
>
> >The number of seeds used is not explicitly stated in the main text (unless I have missed it). The tales in Appendix K report number of trials (which is typically high). I am unsure whether the trial correspond to the number of seeds.
>
> We have clarified in Section 5.1 that the 'trials' reported in the Appendix correspond directly to the number of random seeds used for the experiments.
>
> >I am unsure why the proposed method checks only the empirical variance of the first-order estimate. Typically, zeroth-order estimators also suffer from high variance, which can make the empirical variance less precise. Could the authors clarify that?
>
> The answer to this question depends on the interpretation of what precisely is asked. The IVW method (which is also a component of DDCG) does estimate the empirical variance of the 0th-order gradient as well, and uses this for weighting the gradient estimates (if discontinuities are not detected). However, if you are referring to the $\varepsilon_v$ errorbound on the empirical 1st-order variance estimate that is used in the DDCG test, indeed we do not use such an errorbound for the 0th-order estimate. The first reason for this is that the 0th-order gradient variance is not used in the DDCG discontinuity detection test. In principle, errorbounds could also be added into the IVW calculation to compute some kind of worst or best case weighting based on what the variances could be. However, this is not in the scope of the current work as we are focusing on the bias, not on better variance estimates. In practice we find that simply using the estimated empirical variance without errorbounds works well for variance control.
>
> >A technical note: the baseline subtraction can make REINFORCE biased if it is estimated with the same samples. Look at this example, where b is the average reward (typical baseline subtraction):
>
> In practice, we follow AoBG and use the function estimate at the mean $f(\mu)$, which does not depend on the samples, and is thus unbiased. If one wants to use the samples, the standard approach is to use the Leave-one-out baseline (Parmas et al. ICML2018), which is equivalent to subtracting the mean of the samples and reweighting by $N/(N-1)$, i.e., subtracting the sample mean only biases the magnitude of the gradient, but not the direction. We have added a clarification in Section 3 (under 0th-order estimator) stating that we use a deterministic baseline $b=f(\theta)$ evaluated at the mean parameter. This ensures the estimator remains unbiased, avoiding the bias issues associated with using a sample-dependent baseline.
>
> >Section 4.2 could have a more meaningful title.
>
> We changed the title to “Stepwise Inverse Variance Weighting (IVW-H)”.
>
> Thank you again for the review, and please let us know if there are any further questions or concerns.

---

> > ### Comment · Reviewer_ecYo · 2025-11-24
> > **Satisfied**
> >
> > Dear authors,
> >
> > I am satisfied with the answers and the changes to your manuscript.
> > I was not aware of the leave-one-out baseline subtraction method. Thanks for pointing it out!
> >
> > Best regards.

---

> > > ### Author Response · Authors · 2025-11-26
> > > **Thank you**
> > >
> > > Thank you for the response and the review; we are glad that you are satisfied with the answers.

---

### Official Review · Reviewer_MBBw · 2025-10-30

**Soundness:** 2
**Presentation:** 3
**Contribution:** 3
**Rating:** 6
**Confidence:** 3

**Summary:**

This paper examines the limitations of differentiable physics engines when used to compute policy gradients in reinforcement learning.  In contrast with zero-order methods, such as REINFORCE and its descendants, differentiable physics engines allow direct first-order computation of the policy gradient.  This can reduce the variance of the gradient estimate, since it is less reliant on random sampling.  However, the bias of the gradient estimate may increase when the physical dynamics are not strictly differentiable everywhere - for example, at impulse events. Recent works employ a weighted average of zero- and first-order estimates to get the best of both worlds, but selecting the weights for the averaging is a non-trivial problem.

The present paper explores two methods to set those weights.  The first method (DDCG) uses statistical hypothesis testing to detect when the dynamics are near a discontinuity, in which case the zero-order estimate is used.  The second method (IVW-H) is a variant of a prior technique, "inverse variance weighting," which computes the weights on a per-time-step basis.  The paper includes some theoretical considerations motivating these methods, as well as empirical comparisons with several prior works in several environments - including "toy" environments to aid the analysis as well as some Mujoco continuous control tasks.  On the one hand, the results show that explicit discontinuity detection with DDCG is competitive with prior methods, while being less sensitive to hyper-parameter tuning.  On the other hand, the Mujoco task results show that IVW-H, which does not explicitly detect discontinuities, performs comparably to or better than prior methods.  The paper concludes from the latter results that variance control may be more important than bias control, at least in some Mujoco RL benchmarks.

**Strengths:**

- While I am not fully up to date on differentiable physics simulation, as far as I know the methods introduced in this paper are original.
- The empirical results are rather comprehensive, involving multiple environments of varying complexity and comparison with multiple baselines.
- The empirical results suggest that the new methods are more effective than past work, and likely to be a significant contribution in the sub-field of differentiable physics and first-order policy gradients.
- Aside from certain issues described below, the paper is mostly clear and relatively easy to follow.

**Weaknesses:**

- I have a doubt concerning theoretical soundness.  The DDCG mixing weights are determined via Eq (14) which, according to the paper's notation, appears to use a true variance, not a sample estimate.  But then how is this equation used in practice, if the true variance is not known?
- I also have a doubt about the experimental results and conclusions.  The paper makes a conclusion about the relative importance of variance vs bias control.  However, it does not seem there is any experiment that directly compares the two proposed methods on the same benchmark - DDCG for bias control, IVW-H for variance control.  So, I am not sure the empirical results support the overall conclusion of the paper.  I also wonder why there is no experiment evaluating both methods on the same benchmark.
- In terms of presentation, much important information is pushed to the appendix.  I understand the page limit is tight, but it would be better if the main text were more self-contained.
- Related to the previous two points, this paper appears to explore two orthogonal methods (DDCG, IVW-H), which dilutes the focus of the paper.  If each method were published in a separate paper, each paper would be more focused, and there would more room in the main text for a detailed analysis of each method.
- There are some terms/symbols that are never defined or defined much later in the paper.  In particular:
    - The acronym "AoBG" is never defined, I had to find it in one of the citations.  It should be defined at its first occurrence.
    - In section 4.1, the function $f$ is mentioned abstractly without any specific example of what this function is.  A concrete example is given much later in 5.2.3, but the paper's clarity would benefit if one or more examples are mentioned when $f$ is first defined.

**Questions:**

- Can the authors explain more about Eq 14 - whether the true variance is/must be known for this method to work, why or why not?
- Can the authors justify why they did not evaluate DDCG and IVW-H on the same benchmark?

---

> ### Author Response · Authors · 2025-11-21
> **Response 1/2**
>
> Thank you for the review.
>
> We would first like to clarify the below point from your summary:
>
> >The present paper explores two methods to set those weights. The first method (DDCG) uses statistical hypothesis testing to detect when the dynamics are near a discontinuity, in which case the zero-order estimate is used. The second method (IVW-H) is a variant of a prior technique, "inverse variance weighting," which computes the weights on a per-time-step basis.
>
> We would like to emphasize that DDCG also builds on top of IVW, so it has both bias and variance control. If DDCG detects a discontinuity, it uses the 0th order estimate, but if it does not detect a discontinuity, it uses IVW for variance control. The later IVW-H is a practical implementation of IVW that has good computational efficiency on larger scale MuJoCo-style robotics tasks. IVW-H already worked effectively on these tasks, so further bias detection with DDCG was not necessary on these tasks.
>
> To further clarify, the main motivation of the current work is to create DDCG that improves over the AoBG method (which is also an extension on top of IVW). We achieved this on all tasks from the AoBG paper. However, we also wanted to have MuJoCo-style robotics tasks that were not included in the AoBG paper. However, in these tasks, we found that the “empirical bias” that was claimed to be an issue in the AoBG paper was actually not a major concern. In terms of final results, we improve over AoBG in terms of ease of tuning in all tasks of the AoBG paper, and we improve over GIPPO (another SOTA composite gradient estimation algorithm) in all of the considered MuJoCo-style robotics tasks.

---

> ### Author Response · Authors · 2025-11-21
> **Response 2/2**
>
> **Answers to questions**
>
> >Can the authors explain more about Eq 14 - whether the true variance is/must be known for this method to work, why or why not?
>
> We estimate all quantities in Eq 14 using the empirical sample batch (i.e., the true values are not needed, but can be estimated). For the left-hand side, we added a confidence interval on the gradient variance estimate (which can be obtained from standard statistical techniques). For the function variance on the right-hand side, we could similarly add a confidence interval on the estimate. However, when we tried this in practice, it did not affect the result, so for simplicity we just used the estimate without adding an extra confidence bound term.
>
> >Can the authors justify why they did not evaluate DDCG and IVW-H on the same benchmark?
>
> First, on the AoBG tasks, we **do** include an IVW baseline that is compared against DDCG. The later IVW-H method is essentially a practical implementation of IVW for robotics tasks. There is a key difference in the setup between the AoBG tasks and between the MuJoCo-style robotics tasks that makes it not applicable to use IVW-H in the AoBG tasks. In the AoBG tasks, we are following the setup from the AoBG paper where they sample in parameter space, and then apply the sampled parameter for the full episode. IVW-H, on the other hand, performs sampling in the action-space. Due to performing sampling in the action space, we can follow a similar approach as used in the Total Propagation algorithm (Parmas et al. ICML2018 https://arxiv.org/abs/1902.01240), and combine the gradients in action space, then backpropagate the combined gradient. This has advantages both in terms of estimation efficiency, but also in terms of computational cost, as the actions already include a batch dimension for obtaining the batched gradients for empirical variance estimation (required for using IVW). Consequently, even though the GIPPO paper (https://proceedings.neurips.cc/paper_files/paper/2023/file/1bd8cfc0e4c53869b7f1d0ed4b1e78e1-Paper-Conference.pdf) also included an IVW implementation, they report extremely slow computation speeds, and poor performance. Our IVW and IVW-H implementations are more practical and achieve computation speeds comparable to the standard 1st-order gradient computation. Thus, the main contents of our paper are the introduction of DDCG and the comparison to AoBG. However, as the problems in the AoBG paper are relatively simple, they may not be convincing to some readers. Thus, we added typical robotics tasks that are also used in such composite gradient estimation works. However, we found that simply using IVW (IVW-H) already works effectively on these tasks, and there was no need to use more advanced “empirical-bias” detection methods.
>
> We would also like to emphasize that we beat the state of the art composite gradient estimation methods on both of the benchmarks: AoBG on the “empirical-bias” tasks, and GIPPO on the MuJoCo-style robotics tasks.
>
> Finally, we have updated the paper with tuned AoBG results on the robotics tasks, and it does not outperform the standard IVW, so we do not expect the empirical bias to be an issue here. Similarly, DDCG will not outperform IVW-H on this task (but will have identical performance, as DDCG with c=1.0 will correspond to IVW). However, the assumptions in DDCG do not directly align with the IVW-H implementation, so explaining the adaptation would add much complexity to the paper without improving performance, so we do not include it.
>
> >Related to the previous two points, this paper appears to explore two orthogonal methods (DDCG, IVW-H), which dilutes the focus of the paper. If each method were published in a separate paper, each paper would be more focused, and there would be more room in the main text for a detailed analysis of each method.
>
> We would like to emphasize that IVW is a component of DDCG. DDCG adds a bias detection on top of IVW. However, in the MuJoCo-style tasks, we found that this was unnecessary. In principle we agree that the contents with only the AoBG tasks should be sufficient for the paper, however in practice some readers may be unconvinced, so we added the robotics tasks as well. We would like to further enhance IVW-H with more advanced robotics tasks and further improvements in future work.
>
> Thank you again for the review, and please let us know if there are any further questions or concerns.

---

### Official Review · Reviewer_Dv8Q · 2025-11-01

**Soundness:** 3
**Presentation:** 4
**Contribution:** 4
**Rating:** 6
**Confidence:** 4

**Summary:**

The paper re-examines when analytic policy gradients from differentiable simulators truly help, proposing a lightweight statistical check to safely integrate analytic gradient with model-free gradient estimates via inverse-variance mixing

**Strengths:**

- The proposed check is theoretically sound, clean, simple to verify, and provably more efficient than the existing AoBG approach; consequently, the estimates given by the proposed approach are provably less noisy than AoBG
- Both adequate empirical results and theoretical justifications (except for one point I listed below in weaknesses)
- I enjoyed the presentation, which walks the readers through the problem setup, existing challenges, and their solution smoothly
- I like the experimental setups, especially that the authors design very focused experiments to test the proposed approach

**Weaknesses:**

- Line 147 “This estimator remains unbiased if R is continuous,” — I think this statement is misleading. The first order gradient is only unbiased if the dynamics model, reward function employed are perfect. It is just that in continuous regimes, the dynamics model normally tends to be more accurate. Similarly, “However, when R is discontinuous, the 1st-order estimator can be biased.” — the first-order gradient is almost always biased since there is barely a perfect model.
- It seems that choosing a proper c is task-dependent. Also, assuming a near-quadratic model is restrictive
- Why not also run AoBG in Section 5.3? I think it would be valuable to also see how AoBG performs in these continuous control tasks, especially given that performance gap between AoBG and DDCG is mostly small in previous tasks and that the paper is titled “Does “do differentiable simulators give better policy gradients?” give better policy gradients?”

**Questions:**

- Equation 14 — The proposed check computes variance for the function values at the RHS. How to make sure that the quantity is accurately estimated in the first place? If it needs to be estimated from a sampled batch, then isn’t it running into the same issue and the argument circular?
- Figure 4 (a-b): The estimated alpha values are quite different between AoBG and DDCG in these two cases, but why are the resulting costs almost the same?
- Figure 4 (c): Why the 0th order gradient performs as well as DDCG and AoBG in this case? Does that suggest the evaluation is using too many samples such that the 0th-order gradient estimate is already accurate?

---

> ### Author Response · Authors · 2025-11-21
> **Response 1/2**
>
> Thank you for the review.
>
> >Line 147 “This estimator remains unbiased if R is continuous,” — I think this statement is misleading. The first order gradient is only unbiased if the dynamics model, reward function employed are perfect. It is just that in continuous regimes, the dynamics model normally tends to be more accurate. Similarly, “However, when R is discontinuous, the 1st-order estimator can be biased.” — the first-order gradient is almost always biased since there is barely a perfect model.
>
> This seems to be a misunderstanding regarding what we mean by “unbiased”.
> The “unbiased” refers to the estimator being unbiased for the gradient of the objective inside the simulation, i.e., Eq. (3) holds ($E[g] = \frac{d}{d\theta}E[R]$). Transfer from the simulation to the real environment is done via sim-to-real transfer techniques, and is not related to the gradient estimation. The 0th-order gradient estimator and 1st-order gradient estimator have exactly the same expected value, and both are unbiased for the same objective when $R$ is continuous. We added clarifying remarks regarding this into the paper.
>
> >It seems that choosing a proper c is task-dependent. Also, assuming a near-quadratic model is restrictive
>
> All tasks were run with $c=0.3$ and gave good performance. Also, in appendix H.3, we perform a comprehensive sweep over $c$ values, and find that any $c\in[0.1, 0.9]$ gives good performance on all the tasks. Therefore the method is not sensitive to the choice of $c$. Regarding the “quadraticness” assumption, $c=0$ corresponds to an exactly quadratic function, while $c=1.0$ corresponds to an arbitrary function. Thus, by tuning $c$, the assumption can be loosened based on how smooth we expect the function to be. (Also note that in practice, the assumption is about how close to a quadratic the function is locally, i.e., how smooth the function is. We do not assume global quadratic structure.)
>
> >Why not also run AoBG in Section 5.3? I think it would be valuable to also see how AoBG performs in these continuous control tasks, especially given that the performance gap between AoBG and DDCG is mostly small in previous tasks and that the paper is titled “Does “do differentiable simulators give better policy gradients?” give better policy gradients?”
>
> To address the concern, we evaluated AoBG on the continuous control tasks and included the results in Section 5.3 (Figure 5). The results show that AoBG performs similarly to standard IVW. Furthermore, our sensitivity analysis in Appendix I (Figure 16) reveals that even with extensive hyperparameter tuning, AoBG fails to meaningfully outperform the simpler IVW baseline. This suggests that "empirical bias" is not the dominant bottleneck in these domains, and that the purely variance-based weighting of IVW (and IVW-H) is sufficient.
>
> Regarding the performance difference between AoBG and DDCG: The improvement is mainly in terms of ease of tuning. AoBG has its $\gamma$ parameter tuned separately for each task between $\gamma\in [0.005, 10^8]$, whereas DDCG works effectively for all considered tasks for any fixed $c\in [0.1, 0.9]$ as mentioned in our previous response. The experiments in the main paper were all run with the same fixed $c=0.3$. This greatly improves the practical appeal of DDCG compared to AoBG.
>
> >Equation 14 — The proposed check computes variance for the function values at the RHS. How to make sure that the quantity is accurately estimated in the first place? If it needs to be estimated from a sampled batch, then isn’t it running into the same issue and the argument circular?
>
> Similarly to the variance for the gradient, we can construct a confidence interval for the variance of the function values as well. We tried this, and it did not significantly change the results, so for simplicity we use the estimated variance directly. We added clarifying remarks into the paper.

---

> ### Author Response · Authors · 2025-11-21
> **Response 2/2**
>
> >Figure 4 (a-b): The estimated alpha values are quite different between AoBG and DDCG in these two cases, but why are the resulting costs almost the same?
>
> On 4b) DDCG actually seems a bit better. This is most likely because it makes greater use of the 1st order gradient, which is expected to be more accurate. For clarity 4(a-b) are the same experiment, (a) is run with 100 samples, while (b) is run with 5 samples. On 4a, the sample size is large, so most likely the gradient accuracy is sufficiently large and slight differences do not affect the optimization speed. On 4b, the sample size is small, so the accuracy does have an effect.
>
> >Figure 4 (c): Why the 0th order gradient performs as well as DDCG and AoBG in this case? Does that suggest the evaluation is using too many samples such that the 0th-order gradient estimate is already accurate?
>
> In 4c, the 1st order gradient is highly biased and does not add useful information. For that reason, simply using the 0th order gradient gives good performance. On this task DDCG always detects discontinuities and places all of the weight on the 0th order gradient. To further address your concern, we also reduced the batch size (100 samples) and observed qualitatively similar learning curves: reducing the number of samples makes the trajectories noisier, but DDCG still falls back to the 0th-order gradient and converges to essentially the same performance as the pure 0th-order baseline. Also, please note that we chose the experimental settings (sample sizes, hyperparameters) from the AoBG paper and used the same values, except when we were performing sensitivity and ablation studies.
>
> Thank you again for the review, and please let us know if there are any further questions or concerns.

---

### Official Review · Reviewer_ace9 · 2025-11-01

**Soundness:** 3
**Presentation:** 3
**Contribution:** 3
**Rating:** 4
**Confidence:** 3

**Summary:**

This paper investigates a key challenge in reinforcement learning: how to best calculate policy gradients when using a differentiable simulator. It re-examines the trade-offs between two types of gradient estimators and proposes new methods to improve performance and robustness.

**Strengths:**

- The paper tackles a significant and practical issue in reinforcement learning: how to effectively use differentiable simulators when their gradients are biased by non-smooth dynamics.
- It provides a clear and constructive critique of the AoBG method. It compellingly demonstrates AoBG's key weakness: extreme sensitivity to its hyperparameter which requires extensive, task-specific tuning.
- The paper proposes DDCG, a novel composite estimator. This method is a conceptual advance because its statistical test does not rely on the "notoriously noisy" 0th-order gradient estimator, which improves scalability and sample efficiency compared to AoBG.

**Weaknesses:**

- The paper's experimental validation is split. The proposed DDCG is tested on the "empirical-bias" tasks, while the simpler IVW-H is tested on MuJoCo.
- The conclusion that variance, not bias, is the dominant issue in practice is based on only three MuJoCo tasks.
- The DDCG method largely follows the "test-and-fallback" template of AoBG, with the main contribution being a replacement statistical test. This limits the paper's methodological novelty.

**Questions:**

I have no more questions beyond those mentioned in weaknesses.

---

> ### Author Response · Authors · 2025-11-21
>
> Thank you for the review.
>
> >The paper's experimental validation is split. The proposed DDCG is tested on the "empirical-bias" tasks, while the simpler IVW-H is tested on MuJoCo.
>
> We would like to clarify a few things here. First, on the “empirical-bias” tasks (we will call these the “AoBG tasks”), we **do** include an IVW baseline. The later IVW-H method is essentially a practical implementation of IVW for robotics tasks. There is a key difference in the setup between the AoBG tasks and between the MuJoCo-style robotics tasks that makes it not applicable to use IVW-H in the AoBG tasks. In the AoBG tasks, we are following the setup from the AoBG paper where they sample in parameter space, and then apply the sampled parameter for the full episode. IVW-H, on the other hand, performs sampling in the action-space. Due to performing sampling in the action space, we can follow a similar approach as used in the Total Propagation algorithm (Parmas et al. ICML2018 https://arxiv.org/abs/1902.01240), and combine the gradients in action space, then backpropagate the combined gradient. This has advantages both in terms of estimation efficiency, but also in terms of computational cost, as the actions already include a batch dimension for obtaining the batched gradients for empirical variance estimation (required for using IVW). Consequently, even though the GIPPO paper (https://proceedings.neurips.cc/paper_files/paper/2023/file/1bd8cfc0e4c53869b7f1d0ed4b1e78e1-Paper-Conference.pdf) also included an IVW implementation, they report extremely slow computation speeds, and poor performance. Our IVW and IVW-H implementations are more practical and achieve computation speeds comparable to the standard 1st-order gradient computation. Thus, the main contents of our paper are the introduction of DDCG and the comparison to AoBG. However, as the problems in the AoBG paper are relatively simple, they may not be convincing to some readers. Thus, we added typical robotics tasks that are also used in such composite gradient estimation works. However, we found that simply using IVW (IVW-H) already works effectively on these tasks, and there was no need to use more advanced “empirical-bias” detection methods.
>
> We would also like to emphasize that we beat the state of the art composite gradient estimation methods on both of the benchmarks: AoBG on the “empirical-bias” tasks, and GIPPO on the MuJoCo-style robotics tasks.
>
> Finally, we have updated the paper with tuned AoBG results on the robotics tasks, and it does not outperform the standard IVW, so we do not expect the empirical bias to be an issue here.
>
> >The conclusion that variance, not bias, is the dominant issue in practice is based on only three MuJoCo tasks.
>
> We cannot conclude this in general across all tasks; and indeed, on the AoBG tasks, the empirical bias also matters. However, these robotics tasks are used in several differentiable simulation papers (Son et al. 2023, https://arxiv.org/abs/2312.08710 , Gao et al. 2024 https://proceedings.mlr.press/v235/gao24m.html , Zhang et al. 2023 https://proceedings.mlr.press/v202/zhang23s/zhang23s.pdf), so we believe it is informative to show that the variance is the dominant factor in these tasks. This also explains why we do not run DDCG on these tasks, as the IVW method already works effectively here.
>
> >The DDCG method largely follows the "test-and-fallback" template of AoBG, with the main contribution being a replacement statistical test. This limits the paper's methodological novelty.
>
> Whilst we do build on past work, we do believe that our approach has significant novelty. The statistical test that we propose is conceptually very different to the one used in AoBG. We have discussed this in the “Comparison with AoBG.” paragraph, where we assert that our method has $d$ times better scalability compared to the test used in AoBG. The AoBG method constructs a confidence interval around the 0th-order gradient estimator for its test, thus it relies on this estimate to be accurate; however, 0th-order gradient estimation is notoriously inaccurate, because it is essentially trying to estimate a vector quantity (the gradient) using only scalar evaluations (the function value evaluations). Our method circumvents this issue as the statistical test is not based on the accuracy of 0th-order gradient estimation, but on the accuracy of estimating the function variance (a scalar quantity). Thus, from a conceptual point of view, the test in AoBG is unlikely to be fixable to work efficiently. In the experimental evaluation, indeed our test works effectively even with very small sample sizes like 3 or 10, far lower than the ones used in the original AoBG. While we acknowledge that we build on ideas from prior work, the modification that we propose is not straight-forward and represents a conceptual improvement. We find this sufficiently novel.
>
> Thank you again for the review, and please let us know if there are any further questions or concerns.

---

### Author Response · Authors · 2025-12-03
**Summary of Reviews and Rebuttal Discussion (1/2)**

Dear Area Chair,

Thank you very much for taking on this paper under the unusual circumstances. To help you quickly navigate the reviews and our responses, we provide (i) a brief summary of the review outcome and key strengths, and (ii) an index-style mapping from groups of related issues (weaknesses/questions) to the corresponding rebuttal comments and revised manuscript sections. All substantial changes in the paper are highlighted in red.

We refer to the reviewers as **R1 = ace9 (score 4)**, **R2 = Dv8Q (score 6)**, **R3 = MBBw (score 6)**, **R4 = ecYo (score 8)**. Each weakness/question is denoted **Wk(Ri)** / **Qk(Ri)** following its order in the review.

---

### 1. Review Outcome & Key Strengths

**Scores.** R4: 8, R2: 6, R3: 6 → three reviewers are in favor of acceptance; R1: 4 (marginally below threshold).

**Discussion.** R4 explicitly stated they are satisfied with our answers and the corresponding changes (“Satisfied” comment). The other reviewers could not post follow-ups after the incident, but all of their points are addressed in the rebuttal and incorporated into the revised manuscript. During rebuttal, we also **added AoBG results on the MuJoCo-style robotics tasks**, confirming that AoBG, even with tuning, does not meaningfully improve over our IVW baseline there.

**Strengths repeatedly mentioned across reviews**

- **Important and practical problem.** The paper addresses robust policy gradients with differentiable simulators under non-smooth dynamics, which all reviewers identify as an important and practically relevant RL setting (R1, R2, R3, R4). Our experiments cover all benchmark environments used in the AoBG paper and standard differentiable robotics tasks (MuJoCo-style environments such as Hopper and Ant) that are commonly used in prior composite gradient works.

- **Conceptual contribution of DDCG.** DDCG is recognized as an improved composite estimator over AoBG, using a more efficient function-variance–based test instead of relying on noisy 0th-order gradients (R1, R2, R3). We also show that while AoBG requires task-specific tuning of $\gamma$ over several orders of magnitude (typically $\gamma \in [5 \times 10^{-3}, 10^{8}]$), all our main experiments use a **single fixed $c = 0.3$**, and our sensitivity study (Appendix H.3) shows stable performance: **any $c \in [0.1, 0.9]$ gives good performance on all AoBG tasks**.

- **Bias and variance control in one framework.** Reviewers highlight the combination of bias control (via DDCG’s test) and variance control (via IVW and its practical per-time-step implementation IVW-H for robotics tasks) as a strength of the approach (R2, R3, R4). On the empirical-bias–dominated AoBG tasks, DDCG matches or outperforms AoBG while being much easier to tune, and on the MuJoCo-style tasks our IVW-H implementation outperforms the recent SOTA composite method GIPPO, while the new AoBG experiments added during rebuttal show that even tuned AoBG performs similarly to IVW and does not meaningfully close this gap.

- **Experimental coverage and clarity.** The experiments are described as comprehensive and solid, and after revision the presentation is considered clear and easy to follow (R2, R3, R4), with R4 explicitly confirming satisfaction with the updated version.

---

> ### Author Response · Authors · 2025-12-03
> **Summary of Reviews and Rebuttal Discussion (2/2)**
>
> ### 2. Grouped Issues and Pointers to Our Responses
>
> Below, we group related weaknesses/questions by topic, list the corresponding IDs, and point to where they are handled (rebuttal + sections). Detailed technical content is in the cited rebuttal comments and sections.
>
> ---
>
> **(A) Theory & Eq. (14): unbiasedness, variance estimation, intuition**
>
> **Issues:**
>
> - W1(R2): “unbiased if $R$ is continuous” wording
> - W1(R3): Eq. (14) seems to use true variance
> - W1(R4): Eq. (14) not explained in main text
> - Q1(R2), Q1(R3): how variance in Eq. (14) is estimated; circularity concerns
>
> **Where addressed (rebuttal / paper):**
>
> - Official Comments to **R2 (Response 1/2)**, **R3 (Responses 2/2)**, **R4**
> - **Sec. 3** (definition of unbiasedness), **Sec. 4.1** (Eq. (14) implementation & intuition)
>
> ---
>
> **(B) Experimental coverage, split benchmarks, and “variance vs. bias” conclusion**
>
> **Issues:**
>
> - W1(R1): split evaluation (DDCG on AoBG tasks vs IVW-H on MuJoCo-style tasks)
> - W2(R1): strength of “variance > bias” conclusion (only three MuJoCo-style tasks)
> - W2(R3), Q2(R3): no direct comparison of DDCG and IVW-H on the same benchmark
> - W3(R2): AoBG not run on MuJoCo-style tasks in original version
> - Q2(R2), Q3(R2): interpretation of Fig. 4(a–c)
>
> **Where addressed:**
>
> - Official Comments to **R1**, **R2 (Responses 1/2, 2/2)**, **R3 (Responses 2/2)**
> - **Sec. 5.3 (Fig. 5)**: added AoBG on MuJoCo-style tasks
> - **Appendix I**: AoBG hyperparameter sensitivity
> - Fig. 4 discussion: in rebuttal to **R2 (Response 2/2)**
>
> ---
>
> **(C) Methodological novelty vs. AoBG and overall focus (DDCG vs IVW-H)**
>
> **Issues:**
>
> - W3(R1): DDCG follows AoBG’s test-and-fallback template; limited novelty
> - W4(R3): two seemingly orthogonal methods (DDCG, IVW-H) may dilute focus
>
> **Where addressed:**
>
> - Official Comments to **R1**, **R3 (Response 1/2)**
> - **Sec. 4** (“Comparison with AoBG” paragraph): conceptual differences in the statistical test and sample efficiency
>
> ---
>
> **(D) Hyperparameter $c$ and “quadratic” assumption**
>
> **Issues:**
>
> - W2(R2): task-dependence of $c$; restrictiveness of near-quadratic assumption
>
> **Where addressed:**
>
> - Official Comment to **R2 (Response 1/2)**
> - **Sec. 4.1**: interpretation of $c$ and local smoothness assumption
> - **Appendix H.3**: sensitivity analysis for $c$ (single $c = 0.3$ used across tasks; stable performance over a wide range)
>
> ---
>
> **(E) Presentation, notation, and missing definitions**
>
> **Issues:**
>
> - W3(R3): important information pushed to the appendix; main text should be more self-contained
> - W5(R3): AoBG acronym and function examples not defined at first use
> - W2(R4): AoBG not introduced where it first appears
> - W3(R4): Eqs. (4), (6) look like one-sample estimators
> - W4(R4): compact notation in Eqs. (3)–(6) hides dependencies
> - W5(R4): number of seeds / trials unclear
>
> **Where addressed:**
>
> - Official Comments to **R3 (Response 1/2, 2/2)** and **R4**
> - **Sec. 3**: added “Batch Estimation” paragraph; expanded notation in Eqs. (3)–(6)
> - **Sec. 4.1**: added intuition for Eq. (14) in the main text
> - Main text: AoBG now defined at its first occurrence; more explicit references
> - **Sec. 5.1** & **Appendix K**: clarified that “trials” correspond to random seeds
> - **Sec. 4.2** retitled to “Stepwise Inverse Variance Weighting (IVW-H)”
>
> ---
>
> **(F) DDCG test design (1st-order variance only) and REINFORCE baseline**
>
> **Issues:**
>
> - Q1(R4): why the test checks only 1st-order variance
> - Suggestions(R4): potential bias from baseline subtraction in REINFORCE
>
> **Where addressed:**
>
> - Official Comment to **R4**
> - **Sec. 3**: description of 0th-order estimator and deterministic baseline $b = f(\theta)$ (and mention of leave-one-out baseline)
> - **Sec. 4.1**: role of 1st-order variance in the DDCG test; IVW variance estimates for both 0th- and 1st-order gradients
>
> ---
>
> We hope this index-style mapping from grouped issues to our rebuttal and revised sections helps you quickly locate the relevant material. Thank you again for your time and for taking on this additional responsibility.
>
> Sincerely,
> The authors

---

### Meta-Review · Area_Chair_hNBv · 2026-01-06

**Summary:**

This paper introduces a new estimator that better addresses the challenges of non-smoothness in policy gradient computation, which increases performances on various tasks. Ultimately, most reviewer concerns were quite minor, surrounding presentation issues and thoroughness of experiments. These all seemed to be quite well addressed in the rebuttal, and reviewers expressed their inclination to improve scores should the concerns be addressed.

**Reviewer Concerns:**

Reviewer concerns were generally very minor (presentation, choice of experimental evaluations)., and all concerns were addressed.

**Reviewer Scores:**

Reviewer scores were all positive, and even the "4" score claimed to want to raise the score, but was not able given this year's complications. Specifically, I anticipate that the final scores would have been a 6,6,8,8.

---

### Decision · Program_Chairs · 2026-01-26

Accept (Poster)